# Time-Series Decomposition as a Standalone Task: A Mechanism-Driven Diagnostic Benchmark

Zipeng Wu [1]    Jiani Wei [1]    Shiqiao Zhou [2]    Jiajun Chen [3]    Fabian Spill [1]    James W. Andrews [1]

## Abstract

We benchmark time series decomposition as a standalone evaluation task. While decomposition outputs are widely used to interpret trend and periodic structure, their quality is often assessed informally, and no unified benchmark exists for comparing component recovery under controlled generative mechanisms. We introduce a synthetic evaluation suite with explicit trend and cycle taxonomies, a unified interface covering representative decomposition families, and complementary metrics capturing distinct error modes (shape, phase, and spectral fidelity). Across stationary periodic regimes, STL-family methods are near-ceiling; under non-stationary periodicity (frequency drift, regime switching), fixed-period priors induce phase degradation, while subspace/time-frequency methods better preserve seasonal consistency (adaptive spectral methods may require tuning). We further extend the benchmark with a downstream scientific-discovery track—symbolic regression on decomposed components—showing that a decompose-then-regress pipeline materially improves recoverability and reduces expression complexity, linking decomposition quality to structure discovery. Code, result exports, and the web leaderboard are publicly available through the Hugging Face dataset and leaderboard Space.

## 1. Introduction

Time-series research has long emphasized end-to-end modeling of raw observations for downstream performance, in-cluding forecasting, classification, anomaly detection, imputation, and generation (Wen et al., 2023; Liang et al., 2024). In many real-world settings, the primary objective is to extract and interpret structurally meaningful patterns, such as long-term trend, periodic or quasi-periodic behavior, and residual variation, often with higher priority than improving pointwise predictive accuracy. Time-series decomposition therefore serves as a foundational primitive across domains (Bandara et al., 2025; Dokumentov & Hyndman, 2022; Dudek, 2023): it transforms an opaque sequence into semantically meaningful components, enabling direct reasoning about trend shape and smoothness, the stability and evolution of seasonal patterns, and the nature of residual fluctuations (Wen et al., 2019; 2020). Despite its widespread use, the community still lacks a public, unified, and reusable benchmark that can quantitatively compare decomposition methods under controlled conditions, characterize their failure modes, and support evidence-based method selection.

We view time-series decomposition not merely as a pre-processing heuristic, but as a structured operator applied to a finite-length series $y = \{y_t\}_{t=1}^{N}$, which maps observations to a set of components. Concretely, we consider a decomposition operator $\mathcal{D}$ that produces three core outputs,

$$(T, S, R) = \mathcal{D}(y),$$

where $T$ denotes the trend component capturing low-frequency evolution, $S$ denotes periodic or quasi-periodic structure, and $R$ denotes the residual component aggregating remaining stochastic or irregular variation. This operator view deliberately avoids committing to an additive form: many methods can be interpreted through different combination rules (additive, multiplicative, or more general transformations), while the benchmarking target remains the same-recovering the intended structural components in a way that is consistent with the underlying generative mechanism. Crucially, despite its widespread use, decomposition quality is often judged informally in practice: components that "look plausible" are accepted without systematically verifying whether the recovered trend or periodic structure is close to the underlying patterns of interest. In this work, we evaluate decomposition quality through component recovery: the goal is to reconstruct trend-like and seasonality-like sequences that are similar to the target components under a

---

[1] School of Mathematics, University of Birmingham, Birmingham, B15 2TT, UK [2] School of Computer Science, University of Birmingham, Birmingham, B15 2TT, UK [3] Electronics and Computer Science, University of Southampton, Southampton, SO17 1BJ, UK. Correspondence to: Fabian Spill <f.spill@bham.ac.uk>, James W. Andrews <j.w.andrews@bham.ac.uk>.

*Proceedings of the 43rd International Conference on Machine Learning*, Seoul, South Korea. PMLR 306, 2026. Copyright 2026 by the author(s).

specified generative regime. Importantly, similarity here is multi-faceted rather than pointwise: **a decomposition may preserve overall shape but distort amplitude, align amplitudes but drift in phase, or match time-domain patterns while shifting spectral content**. This motivates complementary metrics that capture distinct error modes, including amplitude mismatch, phase misalignment, frequency-content distortion, and shape disagreement.

Motivated by this perspective, we formalize time-series decomposition as a standalone core learning task and introduce a diagnostic evaluation framework that systematically compares representative methods across a broad taxonomy of controlled generative regimes. Our benchmark covers diverse trend mechanisms and seasonal mechanisms, including settings with non-stationary periodic structure such as frequency drift and regime switching, and provides a unified implementation and evaluation interface for widely used decomposition families (e.g., moving-average filters, STL/MSTL, SSA, EMD/CEEMDAN, VMD, and wavelet-based approaches). Known components enable exact scoring, but the main value is diagnostic: each scenario varies one mechanism at a time to expose inductive-bias mismatch rather than only providing labels. To avoid conclusions driven by a single metric, we evaluate trend and seasonality recovery with complementary criteria that capture both shape alignment and frequency-domain consistency, including Trend $R^2$, Trend DTW distance, Seasonal $R^2$, spectral correlation for $S$, and a max-lag correlation measure that is more robust to phase shifts. Under stationary periodic regimes, classical methods such as STL can be near-optimal; however, under non-stationary periodic regimes, fixed-period assumptions induce phase degradation, while subspace/time-frequency methods better preserve seasonal consistency (adaptive spectral methods can be tuning-sensitive), consistent with their inductive biases.

Beyond evaluating decomposition itself, we introduce time-series symbolic regression as a downstream structure-recovery task, and propose a "decompose-then-regress" pipeline. Trend and periodicity are among the most important recurring patterns in time series, yet they are often entangled with irregular fluctuations and stochastic noise (Park et al., 2025). Direct symbolic regression on raw series must jointly search over mixed trend–seasonality expressions under such entanglement, leading to large search spaces and weak identifiability. By first separating trend-like and periodic structure, decomposition acts as a structured filter that reduces nuisance variation and enables component-wise symbolic regression, which better matches the goal of interpreting meaningful temporal patterns. This perspective also highlights a central challenge: uncontrolled noise and pattern mixing can fundamentally limit symbolic recovery unless the underlying components are adequately isolated by decomposition. This paper makes the following contributions:

- **Problem formulation.** We formalize time-series decomposition as a *component-recovery* problem with ground-truth targets $(T, S, R)$ under a controlled taxonomy of trend, cycle, noise, and event mechanisms, including non-stationary periodic regimes (frequency drift and regime switching).

- **Evaluation protocol.** We propose a method-agnostic alignment protocol that maps heterogeneous method outputs to $(\hat{T}, \hat{S}, \hat{R})$, and evaluate recovery with complementary time-domain and frequency-domain metrics that capture distinct error modes (shape mismatch, phase shift, spectral distortion).

- **Diagnostic findings.** We provide regime-dependent capability maps linking method priors (fixed-period, global-subspace, adaptive-spectral) to systematic success/failure patterns, yielding actionable method-selection guidance.

- **Downstream structure recovery.** We introduce symbolic regression on decomposed components as a scientific-discovery probe, showing that decompose-then-regress improves symbolic recoverability while reducing expression complexity.

*Supporting artifact.* Code, configurations, machine-readable result tables, and the lightweight result browser are available through the Hugging Face dataset and leaderboard Space; full reproducibility details are provided in Appendix B.1.

## 2. Metrics and Protocol

This benchmark evaluates decomposition as recovery of semantic components. For each scenario we generate ground-truth components $(T, S, R)$ and compare them against a method's aligned outputs $(\hat{T}, \hat{S}, \hat{R})$ under a deterministic protocol. The camera-ready core protocol uses six scenarios with 50 deterministic draws per scenario at length $N = 512$, yielding 300 generated synthetic series before method evaluation. All metrics are computed per draw and then aggregated across repeated draws to summarize performance by scenario, by difficulty tier, and overall.

### 2.1. Component Alignment and Period Convention

Different decomposition methods expose components in different native forms (e.g., multiple seasonalities, multiple modes, multiple IMFs, or multi-level wavelet details). To enable method-agnostic evaluation, we deterministically map each method's raw outputs into the benchmark triplet $(\hat{T}, \hat{S}, \hat{R})$ using the same alignment rule across all scenarios. The trend estimate $\hat{T}$ is assigned to the method's lowest-frequency component under its native parameterization. The

seasonal estimate $\hat{S}$ is constructed by aggregating components whose dominant frequency content corresponds to the scenario's intended periodic structure. The residual estimate $\hat{R}$ is defined as the remaining unexplained component after assigning $\hat{T}$ and $\hat{S}$.

The benchmark is not designed to study the "unknown period" setting as a primary research question. Accordingly, periods are injected from scenario metadata (the designated base period in the generator configuration). Methods that require explicit periods (STL/MSTL/RobustSTL) receive them directly, while the Moving Average baseline uses the same value as a seasonal window. For SSA/EMD/CEEMDAN/VMD, the primary period is passed only as a *grouping hint* for frequency-based alignment (not as a tuning oracle). This design choice isolates decomposition recovery and failure modes from the separate problem of period estimation; Appendix C reports FFT-estimated periods, deliberate period misspecification, and alternative component-matching checks.

## 2.2. Trend Metrics

We evaluate trend recovery with two complementary criteria: a scale-sensitive fit metric and a shape-sensitive alignment metric. First, we compute the coefficient of determination between $T$ and $\hat{T}$,

$$R_T^2 = 1 - \frac{\sum_{t=1}^{N}(T_t - \hat{T}_t)^2}{\sum_{t=1}^{N}(T_t - \bar{T})^2},$$

where $\bar{T}$ is the mean of $T$. This metric captures amplitude fidelity and penalizes systematic bias, but can be overly harsh when a method recovers the correct shape with mild phase or scaling distortions.

Second, we compute a Dynamic Time Warping distance between $T$ and $\hat{T}$, denoted $\mathrm{DTW}(T, \hat{T})$, which measures shape similarity under monotone time reparameterization. DTW is reported as a distance (lower is better) and complements $R_T^2$ by being less sensitive to local misalignment while still penalizing gross shape disagreement.

## 2.3. Seasonal Metrics

Seasonal recovery is evaluated by frequency-domain consistency and lag-robust similarity. We also report a Seasonal $R^2$ (defined analogously to Trend $R^2$) as a scale-sensitive fit for $S$. First, we compute a spectral correlation between the power spectra of $S$ and $\hat{S}$. Let $P_S(f)$ and $P_{\hat{S}}(f)$ denote the (normalized) power spectral densities over the discrete frequency grid. The spectral correlation is defined as the Pearson correlation between $P_S$ and $P_{\hat{S}}$, measuring whether a method recovers the correct distribution of periodic energy even when time-domain phase is imperfect.

Second, we compute a max-lag correlation to be robust to

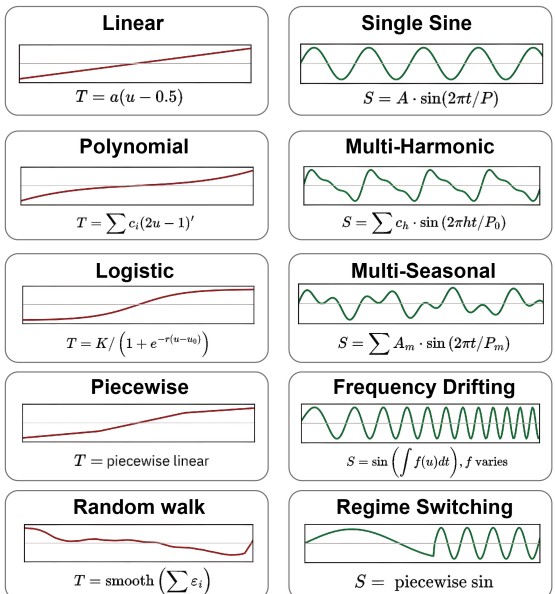

*Figure 1.* **Synthetic Scenario Gallery.** Representative signals across stationary, multi-seasonal, frequency-drifting, and regime-switching + event regimes used in the benchmark.

phase shifts. Let $\rho(\ell)$ be the Pearson correlation between $S_{1:N-\ell}$ and $\hat{S}_{1+\ell:N}$ for lag $\ell$. The max-lag correlation is defined as $\max_{\ell \in \mathcal{L}} \rho(\ell)$ over a fixed lag search window $\mathcal{L}$. This metric detects correct periodic structure when a method's seasonal estimate is phase-shifted relative to the ground truth.

## 2.4. Aggregation and Reporting

Metrics are computed per generated draw (scenario $\times$ deterministic draw id $\times$ method). We summarize results at three levels: (i) **Scenario-level**: mean $\pm$ std across all draws within a scenario; (ii) **Tier-level**: mean $\pm$ std aggregated over scenarios in the same tier; (iii) **Overall**: mean $\pm$ std over all scenarios and draws. For Table 2, we use a tier-balanced display: method means are first computed separately for Tiers 1–3 over valid metric values, stationary columns are the equal-weight average of Tier 1 and Tier 2 means, and non-stationary columns are the Tier 3 means. We also report *coverage* (fraction of runs with valid metrics) to surface failures or missing outputs.

## 3. Methods Under Benchmark

This benchmark compares representative families of time series decomposition methods under a unified component-recovery interface. We evaluate how each family's inductive bias—its implicit prior over trend, periodic structure, and residual—matches or mismatches controlled generative mechanisms. In this section we summarize the evaluated

method families, the components each family tends to favor, and the compatibility expectations that motivate the empirical comparisons.

### 3.1. Method Families and Grouping

We evaluate six broad families that cover common practice in statistics and signal processing. The first family is smoothing-based baselines, represented by Moving Average (Hyndman & Athanasopoulos, 2018), which imposes local constancy and favors low-frequency trend recovery. The second family is seasonal-trend decomposition by local regression, represented by STL (Cleveland et al., 1990) and MSTL (with a robust STL variant) (Bandara et al., 2025; Wen et al., 2019; 2020), which impose smoothness on the trend via LOESS and enforce periodic structure through cycle-subseries smoothing. The third family is subspace decomposition, represented by SSA (Hassani & Thomakos, 2010), which assumes that structured components occupy low-rank Hankel subspaces and separates them via SVD and grouping. The fourth family is empirical mode decomposition, represented by EMD (Huang et al., 1998) and CEEMDAN (Torres et al., 2011; Luukko et al., 2016), which extract intrinsic oscillatory modes through iterative sifting and are intended to adapt to non-stationary amplitude/frequency modulation. The fifth family is variational spectral decomposition, represented by VMD (Dragomiretskiy & Zosso, 2014), which optimizes for a small number of narrow-band modes with learned center frequencies. The sixth family is multi-scale time–frequency decomposition, represented by Wavelet methods (Nason & von Sachs, 1999), which separate components through filter banks with localized support across scales.

**Implementation Notes.** Table 2 uses the default six-family roster (MA, STL, SSA, EMD, VMD, and Wavelet) without hyperparameter search. CEEMDAN is part of the broader sifting-family implementation and appendix/extension checks, but is not an additional Table 2 row. Appendix C reports a bounded common-budget tuning check on a robustness subset; it improves VMD under that budget but is treated as sensitivity evidence rather than a replacement for the default-config leaderboard. For SSA/EMD/VMD in the main run (and CEEMDAN in appendix/extension checks), grouping uses dominant-frequency matching anchored to the injected primary period, providing a fixed evaluation rule whose sensitivity is also checked in the appendix.

### 3.2. Inductive Bias Summary

Each family encodes a different prior over $(T, S, R)$, and these priors determine where a method is expected to succeed and which mechanism violations expose its failure modes. Moving Average favors slowly varying $T$ and treats

deviations as $S$ and $R$ through simple periodic averaging. STL/MSTL favor locally polynomial trends and seasonal structure that is coherent under repeated sampling at seasonal indices, which is strongest under stationary periodicity. SSA favors global subspace separability, performing best when trend and periodic components occupy a small number of Hankel eigentriples and abrupt changes are limited. EMD/CEEMDAN favor locally symmetric oscillatory modes and can adapt to moderate amplitude/frequency modulation, but may suffer from mode mixing in multi-component signals. VMD favors spectrally compact modes around learned center frequencies, making it useful for compact non-stationary cycles but sensitive to $K, \alpha$, and mode-selection choices. Wavelet methods favor multi-scale locality, providing robustness to localized regime changes when relevant frequencies align with selected decomposition scales. Table 1 summarizes these priors as a compatibility matrix used to guide the diagnostics below.

## 4. Results and Analysis

We report results as *capability profiles* rather than a single leaderboard narrative. The benchmark is intended to support two concrete uses: (i) *method selection*—which decomposition family is appropriate under a stated generative regime; and (ii) *method development*—which inductive priors break under specific mechanism violations, and what capabilities a next-generation decomposer should add. Unless otherwise noted, the main results use the default-config, true-period-given protocol, so differences reflect decomposition behavior rather than period estimation; robustness checks for tuning, period information, and alignment are summarized in Appendix C.

### 4.1. Global Landscape: Matched Priors Yield Near-Ceiling Recovery in Stationary Regimes

We first summarize aggregate performance over stationary regimes (Tiers 1–2) versus non-stationary regimes (Tier 3) in Table 2. In stationary settings, families whose priors match the mechanism achieve near-ceiling component recovery. For example, STL-family methods attain high trend fidelity (Trend $R^2 \approx 0.97$) and strong seasonal spectral agreement (Seasonal $\rho_{\text{spec}} \approx 0.98$), while SSA achieves similarly high trend recovery ($R^2 \approx 0.93$) when the signal is well-approximated by a low-rank Hankel structure. This confirms that, under strict periodicity and smooth trends, classical decomposers remain strong baselines and performance differences are often driven by secondary implementation choices (robust weighting, multi-season handling) rather than fundamentally different capability.

In contrast, the non-stationary aggregate highlights a sharp drop in *trend* recovery across families (e.g., STL/SSA trend $R^2$ decline to roughly $0.25/0.20$ on Tier 3, while VMD falls

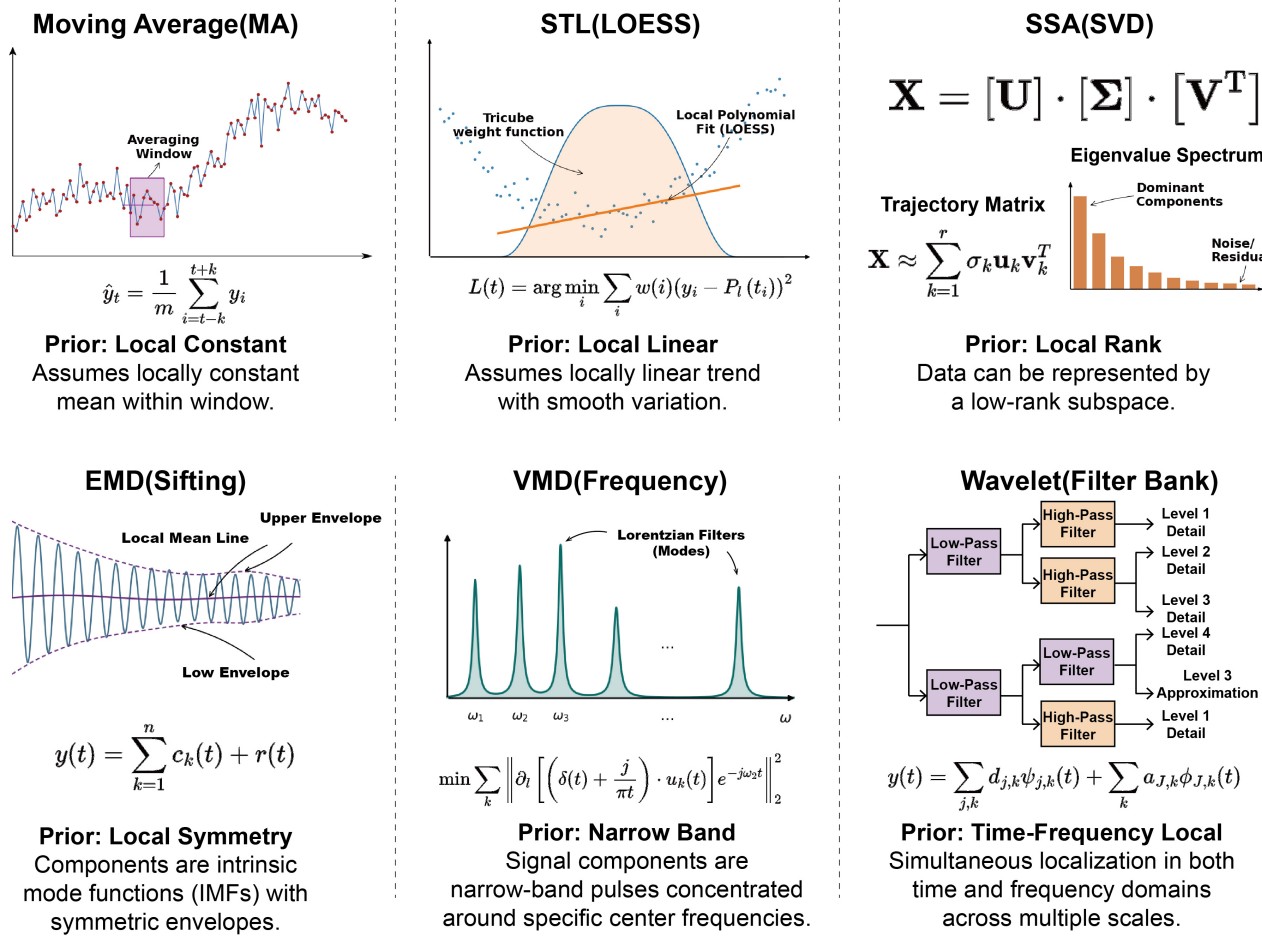

**Figure 2.** **Decomposition Method Collection.** Six families: MA, STL/MSTL, SSA, EMD/CEEMDAN, VMD, and Wavelet. Full formulations are in Appendix G.

below $-1$ under the default setting), while several methods retain high seasonal *spectral* agreement (SSA/Wavelet $\rho_{\text{spec}} \gtrsim 0.98$). This pattern indicates that "non-stationarity" is not a single difficulty axis: it selectively breaks certain priors and often affects trend and seasonality differently. We therefore proceed with mechanism-specific diagnostics rather than treating Tier 3 as a monolithic stress test. The diagnostic values reported below are scenario-level means from the same 50-draw core leaderboard, whereas Table 2 reports tier-level aggregates.

### 4.2. Stationary but Structured Regimes: Multi-Harmonic vs. Multi-Seasonal

Tier 2 contains stationary regimes that remain periodic but are structurally richer, and they expose finer-grained differences that matter for practice.

**Multi-Harmonic Seasonality.** When the seasonal component is a fixed-period signal with multiple harmonics, STL

remains highly effective in the scenario-level six-family core leaderboard: it achieves trend $R^2 = 0.985$ and strong phase-robust seasonal agreement (max-lag correlation $0.945$). This suggests that, when periodicity is coherent and stationary, the LOESS + fixed seasonal-index prior is not only sufficient but close to optimal for component recovery under the protocol.

**Multi-Seasonality.** Multi-seasonal regimes separate trend and seasonal difficulties. In the multi-seasonal scenario-level core leaderboard, trend recovery is tightly clustered across families ($R^2 \approx 0.87$–$0.90$), with SSA slightly highest ($R^2 = 0.900$). Phase-robust seasonal fidelity favors the subspace view in this scenario: SSA has the highest seasonal max-lag correlation and seasonal $R^2$ among the six default families ($0.888/0.773$). This tradeoff is practically important: in multi-seasonal regimes, practitioners may prioritize smooth trend recovery versus a cleaner seasonal subspace for downstream analysis; explicit multi-seasonal LOESS variants are treated in the robustness/extension checks rather

*Table 1.* Compatibility matrix between method-family priors and generating mechanisms.

| Method family | Assumed trend prior | Assumed seasonal prior | Typically compatible regimes | Typical failure regimes |
|---|---|---|---|---|
| Moving Average / smoothing | Local smoothness / local constancy | Fixed-period averaging heuristic | Smooth $T$, simple stationary $S$ | Strong non-stationarity; complex $S$ |
| STL / MSTL (LOESS) | Locally polynomial smoothness | Coherent seasonal index structure (stationary periodicity) | Smooth $T$, stationary $S$ | Frequency drift; regime-switching periodicity |
| SSA (subspace / low-rank Hankel) | Low-rank structured components | Sinusoidal components span low-dimensional subspaces | Stationary sinusoids; moderate complexity | Abrupt regime changes; discontinuities; leakage |
| EMD / CEEMDAN (sifting) | Residual after removing oscillatory modes | Locally symmetric intrinsic modes | AM–FM structure; moderate non-stationarity | Mode mixing in multi-component regimes |
| VMD (variational spectral) | Low-frequency mode separation | Spectrally compact modes around learned $\omega_k$ | Non-stationary $S$ with compact spectrum | Highly broadband $S$; poor K/alpha choices |
| Wavelet (multi-scale) | Coarse-scale approximation | Detail coefficients at selected scales | Local regime effects; multi-scale signals | Energy outside selected bands; heuristic grouping |

*Table 2.* **Global Performance Summary.** Mean Trend $R^2$ and Seasonal Spectral Correlation ($\rho_S$) across stationary (Tiers 1–2) and non-stationary (Tier 3) regimes in the 50-draw camera-ready core protocol. Stationary columns are tier-balanced averages over Tier 1 and Tier 2 means. Bold marks the best value in each column. Classical methods saturate on stationary tasks but degrade on non-stationary trends.

| | Stationary Regimes | | Non-Stationary Regimes | |
|---|---|---|---|---|
| **Method Family** | **Trend** $R^2$ | **Sea.** $\rho_{Spec}$ | **Trend** $R^2$ | **Sea.** $\rho_{Spec}$ |
| Smoothing (MA) | 0.571 | 0.938 | -0.607 | 0.779 |
| LOESS (STL) | **0.965** | **0.980** | **0.247** | 0.973 |
| Subspace (SSA) | 0.929 | 0.876 | 0.195 | **0.987** |
| Sifting (EMD) | 0.936 | 0.867 | 0.119 | 0.964 |
| Spectral (VMD) | 0.077 | 0.185 | -1.488 | 0.046 |
| Wavelet | 0.838 | 0.940 | -0.183 | 0.983 |

than as a separate Table 2 row.

### 4.3. Non-Stationary Diagnostics: Which Priors Break, and What Capability Is Missing

We now isolate two non-stationary mechanisms that correspond to distinct prior violations. Each case is presented as a diagnostic: an empirical signature, the broken assumption, and an actionable design signal.

**Case I: Frequency Drift Violates Fixed-Index Seasonality.** Frequency drift breaks the assumption that seasonality can be indexed by a fixed period ($S_{t+P} = S_t$). For cycle-subseries sampling at a fixed period $P$, the phase increment between samples is approximately

$$\Delta\Phi(n) \approx \frac{2\pi P}{P_0 + \delta u_{k+nP}},$$

which deviates from $2\pi$ when the instantaneous period drifts. Under drift, spectral agreement can remain high (the seasonal energy stays concentrated in a narrow band), yet phase coherence degrades when the model enforces a fixed seasonal index. This is precisely why a phase-robust

metric (seasonal max-lag correlation) is necessary: in the frequency-drift scenario-level core leaderboard, SSA and EMD maintain high max-lag seasonal consistency (0.967 and 0.924), while STL shows noticeably lower phase-robust agreement (0.790) despite retaining high spectral correlation. For a drifting chirp, the effective bandwidth is $\Delta\omega_{\text{chirp}} = 2\pi(1/50 - 1/65)$, while VMD imposes $\Delta\omega_{\text{VMD}} = 1/\sqrt{2\alpha}$, making performance sensitive to the default $K, \alpha$ and mode selection.

*Design signal.* Robust handling of drift requires replacing strict fixed-index seasonality with a tracking-capable prior (time-varying frequency models, time-frequency localized bases, or explicit instantaneous-frequency tracking).

**Case II: Regime Switching and Events Violate Global Smoothness/Low-Rank Structure.** Abrupt trend changes and event bursts violate global smoothness and global separability assumptions. For regime switching, the trajectory matrix is approximately $\mathbf{X} \approx [\mathbf{X}_a \mid \mathbf{X}_b]$, which breaks the low-rank Hankel structure. In the regime-switching scenario-level core leaderboard, all evaluated families exhibit degraded trend recovery (e.g., around $-0.38$ for

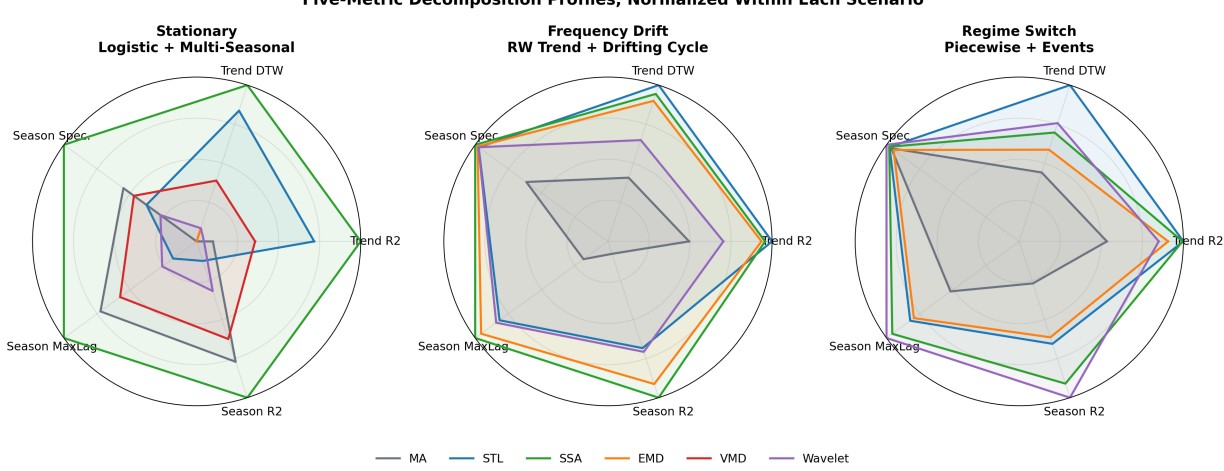

*Figure 3.* **Selected Scenario Radar Profiles.** Three representative regimes (stationary multi-seasonal, frequency drift, regime switching) show method capability signatures across five trend and seasonal metrics, normalized within each scenario with DTW inverted so that larger values indicate better recovery.

STL/SSA and lower for MA/VMD), indicating that global trend recovery is not robust to discontinuities under the current protocol and default configurations. At the same time, seasonality can remain recoverable in that scenario: several time-frequency and subspace/sifting-based families retain high phase-robust seasonal consistency (e.g., wavelet/SSA seasonal max-lag correlations around $\approx 0.88$–$0.91$), suggesting that the seasonal subspace is not the primary bottleneck.

*Design signal.* The missing capability is *locality under structural breaks*: either explicit change-point modeling, segmented trend estimation, or decomposition operators that localize basis functions in time to confine boundary artifacts. Appendix C further checks trimmed-window scoring, showing that boundary sensitivity is largest for VMD and Wavelet but is not a wavelet-only artifact.

### 4.4. From Diagnostics to Action: Method Selection Guidance

In stationary periodic regimes with smooth trends, STL-family methods are reliable defaults and often near-ceiling. In stationary but structured regimes, STL is strong for coherent fixed-period harmonics, while SSA/EMD-like families can yield cleaner seasonal subspaces when phase-robust seasonal fidelity is prioritized. Under frequency drift, selection should prioritize tracking-capable priors (SSA/EMD and time-frequency localized families; CEEMDAN variants are reported in appendix/extension checks). Under regime switching, no tested family reliably recovers the trend component; improving trend recovery requires adding explicit locality or regime awareness.

### 4.5. Scientific Discovery Track: Decomposition Enables Physical Law Recovery

Beyond component recovery, decomposition can serve as a *structure-preserving filter* for downstream law discovery. We evaluate this via Symbolic Regression (SR) on real-world physics cases ($CO_2$ and tides) and a synthetic SR ablation suite, using recent SR backends, PySR (Cranmer, 2023) and ND2 (Yu et al., 2026), to compare Direct SR against Decompose-then-Regress. Because real series rarely provide exact component-level ground truth, additional real-data and semi-synthetic transfer checks are treated as mechanism-aware or proxy evidence rather than exact real-world recovery claims (Appendix C).

**$CO_2$ Physics Validation (Real-World).** On Mauna Loa $CO_2$ (Keeling et al., 2001; Thoning et al., 2025), decomposition makes the seasonal law discoverable: Direct PySR fits the dominant quadratic trend but misses seasonality entirely, whereas decomposition isolates the annual cycle and enables recovery of $\sin(2\pi t)$. In the same experiments, the acceleration term is recovered with near-zero error by STL-family methods, MSTL, VMD, and SSA (all within about 1% relative error on $c_2$; MSTL is below 0.1%). Seasonal amplitude growth (fertilization effect) is detected by most methods, with Wavelet under-detecting relative to others.

**Key Finding ($CO_2$):** Direct SR fits the dominant quadratic trend ($R_T^2 = 1.0$) but completely fails to recover the annual seasonal law (Seasonal $R^2 = 0.0$)—the algorithm treats the 12-month cycle as noise. The decompose-then-regress pipeline changes this outcome: by first isolating the seasonal component, SR discovers a sinusoidal expression with frequency $\omega \approx 2\pi$ rad/year, matching the known annual carbon cycle. VMD+PySR achieves the highest seasonal

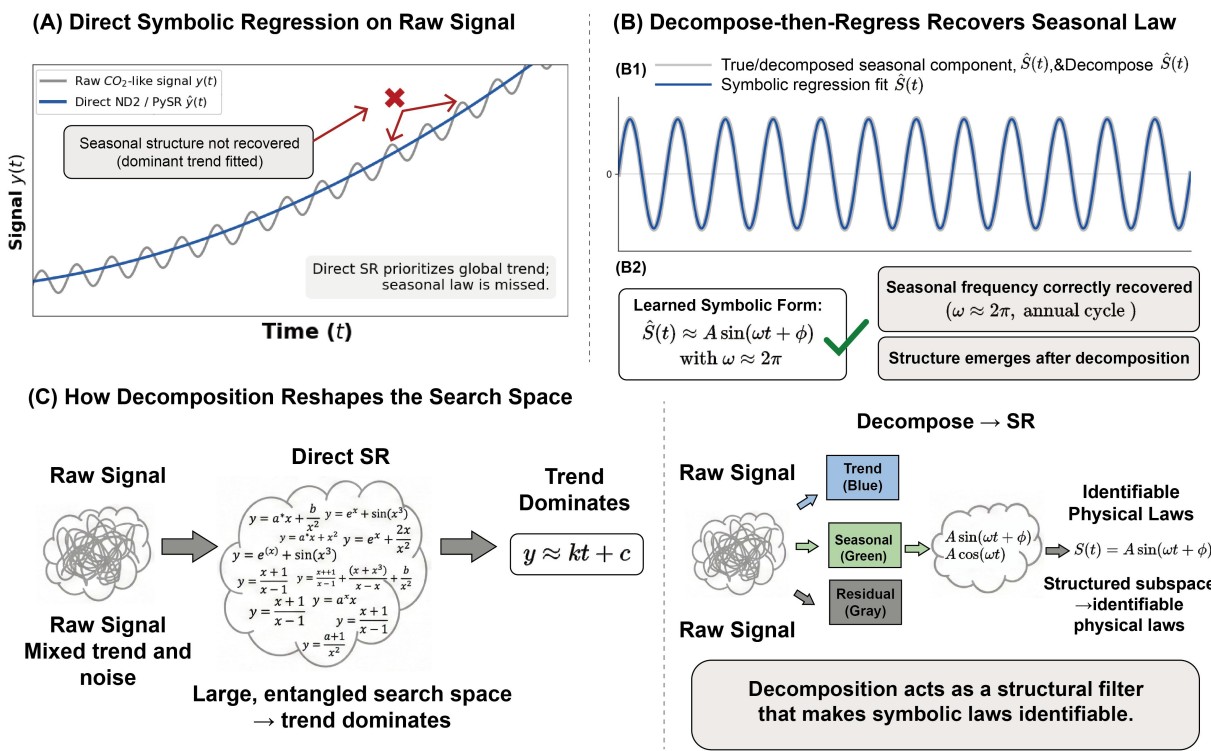

*Figure 4.* **Time-Series Structure Recovery.** Decompose-then-regress narrows the symbolic search space, avoiding trend-dominated local minima and improving recovery of seasonal laws.

*Table 3.* **Seasonal Law Recovery on Mauna Loa $CO_2$.** Direct SR achieves near-perfect trend fit but treats seasonality as noise and fails to extract a seasonal signal. Decomposition isolates the seasonal subspace, enabling recovery of a sinusoidal annual cycle ($\omega \approx 2\pi$) when present.

| Method | Trend | Seasonal | Formula |
|---|---|---|---|
| Direct PySR | 1.000 | 0.000 | (none) |
| STL + PySR | 1.000 | 0.897 | $\sin(2\pi t)$ |
| CEEMDAN + PySR | 1.000 | 0.897 | $\sin(2\pi t)$ |
| VMD + PySR | 1.000 | **0.989** | $\sin(2\pi t)$ |

recovery ($R^2 = 0.989$), followed by STL and CEEMDAN ($R^2 \approx 0.90$).

**Tidal Case Study (Real-World Limitation).** On the tidal dataset (M2/S2 constituents separated by only 3.4% in frequency), the downstream SR setting remains difficult: generic decompose-then-regress does not reliably recover separate symbolic M2/S2 formulas. Direct PySR performs poorly; Decomp+PySR improves seasonal $R^2$ to the 0.46–0.60 range, but still does not resolve M2 vs S2 at the formula level. This is a clear negative result: generic decomposition is not a substitute for domain-specific harmonic analysis when frequencies are extremely close.

**Synthetic SR Ablation (ND2).** On synthetic interference regimes, decomposition improves *trend* discovery but seasonal formula recovery remains challenging. For linear-plus-sine, SSA+ND2 improves the final reconstruction ($R^2 \approx 0.31$) versus near-zero for Direct ND2, while STL+ND2 improves trend SR ($R^2 \approx 0.85$) but still struggles on seasonal structure. For linear-plus-multi-harmonic, STL+ND2 boosts trend SR to $\approx 0.91$ (vs. $\approx 0.55$ direct), yet seasonal SR remains low ($\approx 0.03$), indicating that multi-harmonic seasonality remains a bottleneck for current SR backends.

Taken together, these results show that decomposition acts as a *mechanism-identifiable filter* for SR—it simplifies the search space and improves recovery of physically meaningful structure—but it does not eliminate hard cases such as close-frequency separation or multi-harmonic seasonality. Full experimental details are provided in the appendix.

## 5. Discussion and Conclusion

This benchmark elevates decomposition to an independently evaluated task centered on component recovery. Our results demonstrate that method choice inevitably imposes structural priors: fixed-period assumptions (STL) succeed in stationary regimes but degrade under drift, while

wavelet/subspace methods better preserve seasonal consistency and VMD is parameter-sensitive under default settings.

By isolating failure modes into a compact taxonomy—frequency drift versus fixed-period priors, and regime switching versus global bases—we provide a theory-consistent guide for method selection. Furthermore, we bridge decomposition to downstream discovery, showing that "decompose-then-regress" acts as a structural filter that significantly improves the recoverability of physical laws. Semi-synthetic transfer checks preserve the mechanism-level picture but make several method-level statements more conservative; in particular, real low-frequency backgrounds can induce CEEMDAN mode mixing that preserves frequency agreement while damaging amplitude and shape recovery.

Limitations include the use of exogenous event drivers rather than explicit event decomposition, and the focus on component recovery over forecasting. The primary leaderboard focuses on standalone decomposition operators; future benchmark additions should be treated as external artifact extensions rather than changes to the frozen six-family paper summary. Appendix C also includes a synthetic MSSA pilot for multi-sequence panels, while full real-panel benchmarking remains future work. Future work should extend this protocol to unknown-period estimation and explicit change-point localization. Taken together, these assets provide a reproducible foundation for evidence-based decomposition research.

**Reproducibility**: We provide a unified experimental API and machine-readable result exports to support future extensions. The public artifact contains code, configs, and full result tables.

## Reproducibility and Data Availability

Code, configurations, and experimental results are available through the public Hugging Face dataset and leaderboard artifacts listed above, with full reproducibility details in Appendix B.1. The main benchmark and SR experiments were run on a single A100 compute node; additional robustness and transfer checks include their run metadata in the artifact.

## Acknowledgments

F.S. was supported by a UKRI Future Leaders Fellowship, grant number [MR/T043571/1].

## Impact Statement

This paper presents work whose goal is to advance the field of Machine Learning. There are many potential societal consequences of our work, none which we feel must be specifically highlighted here.

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

# A. Full Synthetic Generative Mechanisms and Protocol Details

We now detail the exact mathematical equations used to generate synthetic time series in the `generator.py` module.

## A.1. Time Axis Definition

$$t = [0, 1, 2, \ldots, N-1] \cdot \Delta t, \qquad (1)$$

$$u = [0, \frac{1}{N-1}, \frac{2}{N-1}, \ldots, 1]. \qquad (2)$$

## A.2. Trend Functions

### A.2.1. LINEAR TREND

$$T_t = \text{scale}\left(a \cdot (u_t - 0.5) + b\right), \qquad (3)$$

where $a \sim \mathcal{U}(-2, 2)$, $b = 0$, and $\text{scale}(\cdot)$ centers and normalizes to target amplitude.

### A.2.2. POLYNOMIAL TREND

For degree $d$ with coefficients $\boldsymbol{c} = [c_0, c_1, \ldots, c_d]$:

$$T_t = \text{scale}\left(\sum_{i=0}^{d} c_i (2u_t - 1)^i\right). \tag{4}$$

### A.2.3. LOGISTIC GROWTH TREND

$$T_t = \text{scale}\left(\frac{K}{1 + \exp(-r(u_t - u_0))}\right). \tag{5}$$

**Default parameters**: $K = 1.0$, $r = 10.0$, $u_0 = 0.5$.

**Code** (`generator.py`, lines 136-147):

### A.2.4. EXPONENTIAL TREND

$$T_t = \text{scale}\left(\alpha \exp\left(\beta(u_t - 0.5)\right)\right). \tag{6}$$

**Default parameters**: $\alpha = 0.5$, $\beta = 2.0$.

### A.2.5. PIECEWISE LINEAR TREND

Let $\{\xi_1, \xi_2, \ldots, \xi_m\} \subset (0, 1)$ be breakpoints. Let $\{a_0, a_1, \ldots, a_m\}$ be slopes.

For $u_t \in [\xi_{k-1}, \xi_k)$:

$$T_t = T_{\xi_{k-1}} + a_k(u_t - \xi_{k-1}), \tag{7}$$

where $T_0 = 0$ and the function is continuous at each breakpoint.

### A.2.6. RANDOM WALK WITH SMOOTHING

$$W_t = \sum_{i=1}^{t} \varepsilon_i, \quad \varepsilon_i \sim \mathcal{N}(0, \sigma_{step}^2), \tag{8}$$

$$T_t = \text{scale}\left(\text{MA}_w(W_t)\right), \tag{9}$$

where $\text{MA}_w$ is a moving average with window $w$.

### A.3. Seasonal (Cycle) Functions

### A.3.1. SINGLE SINUSOID

$$S_t = A \sin\left(\frac{2\pi t}{P} + \phi\right). \tag{10}$$

### A.3.2. MULTI-HARMONIC

With base period $P_0$ and $H$ harmonics:

$$S_t = \sum_{h=1}^{H} c_h \sin\left(\frac{2\pi h t}{P_0}\right). \tag{11}$$

### A.3.3. MULTI-SEASONAL

$$S_t = \sum_{m=1}^{M} A_m \sin\left(\frac{2\pi t}{P_m}\right), \tag{12}$$

with periods $\{P_1, P_2, \ldots, P_M\}$ and amplitudes $\{A_1, A_2, \ldots, A_M\}$.

### A.3.4. SAWTOOTH WAVE

$$S_t = A\left(2\left(\frac{t}{P} \mod 1\right) - 1\right). \tag{13}$$

### A.3.5. SQUARE WAVE

$$S_t = A \cdot \text{sign}\left(\sin\left(\frac{2\pi t}{P}\right)\right). \tag{14}$$

### A.3.6. AMPLITUDE-MODULATED CYCLE

Let the amplitude vary linearly over time:

$$S_t = (A_0 + (A_1 - A_0)u_t)\sin\left(\frac{2\pi t}{P}\right), \tag{15}$$

where $A_0$ and $A_1$ are the initial and final amplitudes.

**Default parameters**: $A_0 = 0.5$, $A_1 = 1.5$.

### A.3.7. FREQUENCY-DRIFTING (CHIRP)

Let the instantaneous period vary linearly:

$$P(u) = P_0 + \delta \cdot u. \tag{16}$$

Instantaneous frequency:

$$f(u) = \frac{1}{P(u)}. \tag{17}$$

Phase is the integral of frequency:

$$\Phi_t = 2\pi \sum_{i=1}^{t} f(u_i)\Delta t. \tag{18}$$

Seasonal component:

$$S_t = A \sin(\Phi_t). \tag{19}$$

**Code** (`generator.py`, lines 282-295):

### A.3.8. REGIME-SWITCHING CYCLE

Let $\tau \in (0, 1)$ be the regime switch point

$$S_t = \begin{cases} A_a \sin\left(\frac{2\pi t}{P_a}\right) & u_t < \tau \\ A_b \sin\left(\frac{2\pi t}{P_b}\right) & u_t \geq \tau \end{cases}. \tag{20}$$

## A.4. Noise Models

### A.4.1. WHITE NOISE

$$\varepsilon_t \sim \mathcal{N}(0, \sigma^2) \,. \tag{21}$$

### A.4.2. AR(1) NOISE

$$\varepsilon_t = \phi \varepsilon_{t-1} + \eta_t, \quad \eta_t \sim \mathcal{N}(0, \sigma^2) \,. \tag{22}$$

### A.4.3. ARMA(1,1) NOISE

$$\varepsilon_t = \phi \varepsilon_{t-1} + \eta_t + \theta \eta_{t-1} \,. \tag{23}$$

### A.4.4. GARCH-LIKE VOLATILITY

$$\sigma_t^2 = \omega + \alpha \varepsilon_{t-1}^2 + \beta \sigma_{t-1}^2 \,, \tag{24}$$

$$\varepsilon_t = \sigma_t \cdot z_t, \quad z_t \sim \mathcal{N}(0, 1) \,. \tag{25}$$

### A.4.5. BURSTY NOISE

Background white noise with intermittent high-variance bursts:

$$\varepsilon_t = \eta_t + \sum_{b=1}^{B} \mathbb{I}_{t \in [t_b, t_b + L_b)} \cdot \zeta_t^{(b)} \,, \tag{26}$$

where $\eta_t \sim \mathcal{N}(0, \sigma^2)$, $\zeta_t^{(b)} \sim \mathcal{N}(0, \sigma_{\text{burst}}^2)$, and $B$ bursts of length $L_b$ are placed at random locations.

## A.5. Event Models

### A.5.1. LEVEL SHIFTS

At times $\{t_1, t_2, \ldots\}$:

$$E_t = \sum_{k=1}^{m} c_k \cdot \mathbb{I}_{t \geq t_k} \,. \tag{27}$$

### A.5.2. SPIKES

At times $\{t_1, t_2, \ldots\}$:

$$E_t = \sum_{k} \pm s_k \cdot \delta_{t, t_k} \,, \tag{28}$$

where $s_k$ is the spike magnitude.

### A.5.3. MIXED EVENTS

Combines level shifts and spikes:

$$E_t = \sum_{k=1}^{m} c_k \cdot \mathbb{I}_{t \geq t_k} + \sum_{j} \pm s_j \cdot \delta_{t, t_j} \,. \tag{29}$$

## A.6. SNR Scaling

Noise is scaled to achieve a target Signal-to-Noise Ratio:

$$\text{SNR} = \frac{\text{RMS}(T + S + E)}{\text{RMS}(\varepsilon)} \,. \tag{30}$$

Targets: High = 5.0, Medium = 2.0, Low = 1.0.

# B. Detailed Evaluation Metrics

We evaluate trend recovery with two complementary criteria: a scale-sensitive fit metric and a shape-sensitive alignment metric.

First, we compute the coefficient of determination between $T$ and $\hat{T}$ as follows:

$$R_T^2 = 1 - \frac{\sum_{t=1}^{N}(T_t - \hat{T}_t)^2}{\sum_{t=1}^{N}(T_t - \bar{T})^2},$$

where $\bar{T}$ is the mean of $T$. This metric captures amplitude fidelity and penalizes systematic bias.

Seasonal $R^2$ is computed analogously:

$$R_S^2 = 1 - \frac{\sum_{t=1}^{N}(S_t - \hat{S}_t)^2}{\sum_{t=1}^{N}(S_t - \bar{S})^2}.$$

Second, we compute a Dynamic Time Warping distance between $T$ and $\hat{T}$, denoted $\text{DTW}(T, \hat{T})$, which measures shape similarity under monotone time reparameterization. DTW is reported as a distance (lower is better) and complements $R_T^2$ by being less sensitive to local misalignment while still penalizing gross shape disagreement.

Seasonal recovery is evaluated by frequency-domain consistency and lag-robust similarity. First, we compute a spectral correlation between the power spectra of $S$ and $\hat{S}$. Let $P_S(f)$ and $P_{\hat{S}}(f)$ denote the (normalized) power spectral densities. The spectral correlation is defined as the Pearson correlation between $P_S$ and $P_{\hat{S}}$, measuring whether a method recovers the correct distribution of periodic energy even when time-domain phase is imperfect.

Second, we compute a max-lag correlation to be robust to phase shifts. Let $\rho(\ell)$ be the Pearson correlation between $S_{1:N-\ell}$ and $\hat{S}_{1+\ell:N}$ for lag $\ell$. The max-lag correlation is defined as $\max_{\ell \in \mathcal{L}} \rho(\ell)$ over a fixed lag search window $\mathcal{L}$. This metric detects correct periodic structure when a method's seasonal estimate is phase-shifted relative to the ground truth.

## B.1. Reproducibility and Data Availability Details

Code, configurations, and experimental results are available through the public Hugging Face artifacts listed in the main

text. The artifact provides repository-style source code, a one-command runner for the primary 6-scenario, 50-draw, six-family core leaderboard used for Table 2 and Figure 3, exact CSV versions of the camera-ready tables, environment snapshots (Git commit hash, package versions, and platform metadata exported automatically with each run), and an extensible API with documented interfaces for adding new scenarios, methods, and metrics.

The main benchmark and SR experiments were run on a single A100 compute node. Additional robustness and transfer checks in Appendix C are released as exact table summaries with available source summaries and run metadata; they are rebuttal-stage sensitivity checks rather than replacements for the one-command primary leaderboard. For CPU-only reproduction, the 6-scenario, 50-draw, six-family core leaderboard requires minutes to tens of minutes on a standard workstation (Intel i7 or equivalent), depending on the available numerical backend. The $CO_2$ symbolic regression experiments add approximately 2 hours (ND2 backend) to 8 hours (PySR backend with default budgets). All results are deterministic given fixed random seeds.

**Datasets**: The Mauna Loa $CO_2$ dataset (Thoning et al., 2025) and NOAA CO-OPS tides and harmonic-constituent data (NOAA Center for Operational Oceanographic Products and Services, n.d.) are publicly available from NOAA; synthetic scenarios are generated on-the-fly from parameterized distributions (Appendix A).

# C. Additional Robustness, Transfer, and Extension Checks

This appendix summarizes additional robustness, transfer, and extension checks. These checks are not used to replace the primary leaderboard in the main text; instead, they quantify sensitivity to tuning, period information, component alignment, boundary scoring, real-background transfer, and multi-sequence structure. The rounded values reported below are mirrored in the public artifact as camera-ready table CSVs.

## C.1. Bounded Tuning

The main paper reports default configurations to compare method-family priors under a fixed protocol. To test whether default settings unfairly penalize parameter-sensitive families, we also ran bounded common-budget tuning on a computationally bounded 5-scenario, 24-draw, length-512 robustness subset. The search spaces were fixed before evaluation: VMD used $K \in \{2, 3, 4, 5, 6\}$, $\alpha \in \{200, 500, 1000, 2000, 5000\}$, and init $\in \{1, 2\}$; CEEMDAN used trials $\in \{50, 100\}$ and noise_scale $\in \{0.05, 0.1, 0.2\}$; SSA used grouping_rule $\in \{\text{current\_period\_aware}, \text{spectral\_auto}\}$, rank $= 10$, and

windows $\{N/4, N/3, N/2, 2N/3, P, 2P, 4P\}$.

*Table 4.* **Bounded tuning check.** On the rebuttal-stage robustness subset, common-budget tuning improves VMD but does not overturn the leading summary order.

| Setting | Aggregate order (best → worst) |
|---|---|
| Default | SSA > CEEMDAN > MA > STL > Wavelet > VMD |
| Tuned | SSA > CEEMDAN > MA > VMD > STL > Wavelet |

## C.2. Period and Alignment Robustness

The primary protocol provides the true base period to period-requiring methods to isolate decomposition behavior from period estimation. We therefore ran two robustness probes. First, we estimated the period by FFT on the first 60% of each series and swept explicit period misspecification up to $+20\%$. Second, for multi-component methods, we compared injected-period matching with unsupervised peak matching and truth-guided subset matching. The latter uses ground truth only to select the best component assignment for scoring; the decomposition outputs are unchanged.

*Table 5.* **Period robustness for STL-family methods.** Reasonable FFT estimates remain competitive, while larger misspecification sharply reduces seasonal recovery.

| Method | Setting | Trend $R^2$ | Sea. spec. | Sea. max-lag |
|---|---|---|---|---|
| STL | True | 0.954 | 0.913 | 0.889 |
| STL | FFT | 0.884 | 0.823 | 0.798 |
| STL | $+20\%$ | 0.928 | 0.512 | 0.405 |
| MSTL | True | 0.987 | 0.958 | 0.913 |
| MSTL | FFT | 0.886 | 0.815 | 0.790 |
| MSTL | $+20\%$ | 0.940 | 0.467 | 0.338 |

*Table 6.* **Alignment robustness.** Component matching affects multi-component families more than STL/MSTL, so rankings should be read as protocol-dependent diagnostics rather than universal method orderings.

| Method | Injected | Unsupervised | Truth-guided |
|---|---|---|---|
| MSTL | 0.757 | 0.757 | 0.757 |
| STL | 0.756 | 0.756 | 0.756 |
| SSA | 0.730 | 0.763 | 0.862 |
| CEEMDAN | 0.727 | 0.355 | 0.867 |
| Wavelet | 0.636 | 0.292 | 0.720 |
| VMD | -0.283 | 0.274 | 0.559 |

## C.3. Boundary Sensitivity

Boundary handling can affect finite-length decomposition metrics. We therefore compared full-window scoring with trimmed-window scoring over lengths $128/256/512/1024$, reporting the mean absolute suite-score delta as a sensitivity diagnostic rather than a primary ranking metric.

## C.4. Real-Data Proxy and Semi-Synthetic Transfer

Real time series rarely contain exact component-level ground truth for $T, S, R$. We therefore separate mechanism-

*Table 7.* **Boundary sensitivity.** Edge handling matters most for VMD and Wavelet, but the effect is not wavelet-only.

| Method | Mean abs. suite-score delta |
|--------|------------------------------|
| VMD | 1.097 |
| Wavelet | 0.734 |
| SSA | 0.311 |
| CEEMDAN | 0.230 |
| STL | 0.224 |
| MSTL | 0.209 |

aware cases from broader proxy/stability checks. On mechanism-aware datasets, SSA achieves $CO_2$ seasonal/trend $R^2 = 0.993/1.000$, while MSTL achieves tidal seasonal $R^2 = 0.956$, target-frequency coverage $1.000$, and M2/S2 separation success $1.000$. On a broader panel ($CH_4$, NDVI, GPCC precipitation, Arctic sea ice, sunspots, and QBO), the evaluation uses band plausibility, resampling stability, and temporal spectrum overlap rather than exact recovery.

*Table 8.* **Broad real-data proxy panel.** Values summarize proxy/stability evidence, not exact component recovery.

| Method | Band plaus. | Resamp. stab. | Spec. overlap |
|--------|-------------|---------------|---------------|
| SSA | 0.967 | 0.651 | 0.950 |
| MSTL | 0.918 | 0.652 | 0.951 |
| STL | 0.910 | 0.639 | 0.947 |
| VMD | 0.792 | 0.570 | 0.882 |
| CEEMDAN | 0.782 | 0.557 | 0.903 |
| Wavelet | 0.172 | 0.666 | 0.924 |

We also injected known mechanisms into six real monthly backgrounds using three mechanisms, two background scales, and eight windows per setting at length 480. The aggregate mean-rank order was SSA > MA > STL > VMD > CEEMDAN > Wavelet. This camera-ready transfer snapshot preserves the broad mechanism-level picture but refines method-level statements. In particular, CEEMDAN does not lose the cycle entirely: its mean spectral correlation remains $0.769$ and max-lag correlation $0.704$, but seasonal $R^2$ drops to $-0.232$ because real low-frequency backgrounds induce mode mixing. At background scale $0.5$, CEEMDAN seasonal $R^2$ remains $0.57$–$0.68$; at scale $1.0$, it becomes strongly negative across mechanisms. The separate 22-method semi-synthetic release in the public artifact is a later benchmark extension and should be read separately from this paper snapshot.

## C.5. Multi-Sequence Extension

The benchmark can also support correlated-panel studies. In a synthetic multi-sequence pilot with shared annual/semiannual, drifting-frequency, and regime-shift-with-local-shocks families at $C = 8$ and $C = 32$, MSSA achieves mean channel trend/seasonal $R^2 = 0.989/0.970$, seasonal covariance error $0.035$, and shared-loading cor-

relation $0.999$. In the hardest shared drifting-frequency panel with 32 channels, MSSA reaches $0.991/1.000$, compared with STL $0.755/0.963$ and single-sequence SSA $0.875/0.996$. We treat this as initial synthetic evidence for multi-sequence extension rather than a complete real-panel benchmark.

Finally, the public artifact is structured to support future benchmark extensions. Such additions are useful for benchmark growth, but they are not used to alter the primary standalone-decomposition leaderboard reported in the main text.

## D. Full Theoretical Analyses and Derivations

### D.1. Analysis 1: STL's Fixed-Period Assumption vs. Frequency Drift

**Scenario**: `rw_trend_freq_drifting_cycle`

**Data Generation** (from `generator.py`):

$$S_t^{\text{gen}} = \sin\left(2\pi \sum_{i=1}^{t} f(u_i)\,\Delta t\right), \quad f(u) = \frac{1}{P_0 + \delta u}, \tag{31}$$

with $u \in [0, 1]$, $\Delta t = 1$, $P_0 = 50$, $\delta = 15$. Thus the instantaneous period drifts from 50 to 65 over the series (length $N = 512$).

**STL Assumption (as run in the benchmark)**:

$$S_t^{\text{stl}} = S_{t+P}^{\text{stl}}, \tag{32}$$

$\forall t$ with fixed $P = 50$ (true period given by the benchmark).

**Mismatch Analysis**:

The phase of the generated signal is:

$$\Phi(t) = 2\pi \sum_{i=1}^{t} \frac{1}{P_0 + \delta u_i}\,\Delta t. \tag{33}$$

The STL cycle-subseries for index $k$ samples points at times $\{k, k + P, k + 2P, \ldots\}$. The phase difference between consecutive samples is:

$$\Delta\Phi = \Phi(k + nP) - \Phi(k + (n-1)P). \tag{34}$$

For a stationary signal, $\Delta\Phi = 2\pi$ (exactly one cycle). For our drifting signal:

$$\Delta\Phi(n) \approx 2\pi \sum_{t=k+(n-1)P}^{k+nP} \frac{1}{P_0 + \delta u_t}\,\Delta t. \tag{35}$$

Using a local approximation:

$$\Delta\Phi(n) \approx \frac{2\pi P}{P_0 + \delta\, u_{k+nP}}. \tag{36}$$

When $P \neq P(t)$, the accumulated phase error is:

$$\text{Error} = \sum_n |\Delta\Phi(n) - 2\pi| . \tag{37}$$

For $\delta = 15$, $P = 50$, $N = 512$:
- At $u \approx 0$: $P_{true} \approx 50$, $\Delta\Phi \approx 2\pi$.
- At $u \approx 1$: $P_{true} \approx 65$, $\Delta\Phi \approx 2\pi \cdot 50/65 \approx 0.77 \cdot 2\pi$.

Neither matches $2\pi$. The cycle-subseries contains values from **different phases** of the sinusoid, and their average is corrupted.

**Predicted Result**: STL degrades as drift accumulates, but not catastrophically under the true period given $P = 50$.

**Observed Result** (from the 50-draw camera-ready run): $S\_r2 \approx 0.61$, $T\_r2 \approx 0.87$, and spectral correlation $\rho_S \approx 0.97$. STL is therefore only moderately degraded in this setting (not catastrophic failure).

### D.2. Analysis 2: SSA's Hankel Matrix and Spectral Leakage

**Scenario**: `piecewise_trend_regime_cycle_with_events`

**Data Generation** (from `scenarios.py`):

$$S_t = \begin{cases} \sin(\omega_a t) & t < \tau \\ 0.6\sin(\omega_b t) & t \geq \tau \end{cases} , \tag{38}$$

where $\omega_a = 2\pi/40$, $\omega_b = 2\pi/65$, $\tau = 0.55N$, with additional trend breaks and mixed events (shifts + spikes) in the full scenario.

**SSA Theoretical Framework**:

For a pure sinusoid $\sin(\omega t)$ of period $P$, the trajectory matrix $\mathbf{X}$ has rank 2 (corresponding to $\sin$ and $\cos$ components). The eigenvalues are:

$$\lambda_1 = \lambda_2 = \frac{N \cdot A^2}{2} . \tag{39}$$

For a signal with **two distinct frequencies** $\omega_a$ and $\omega_b$, the trajectory matrix should have rank 4.

However, if the frequencies switch **abruptly** at $\tau$, the Hankel structure is broken. The matrix is approximately:

$$\mathbf{X} \approx \begin{bmatrix} \mathbf{X}_a & \mathbf{X}_b \end{bmatrix} , \tag{40}$$

where $\mathbf{X}_a$ corresponds to $t < \tau$ and $\mathbf{X}_b$ to $t \geq \tau$.

The SVD of this concatenation does **not** cleanly separate into eigentriples for $\omega_a$ and $\omega_b$. Instead, the singular vectors $\mathbf{u}_i$ become **mixtures** of both frequencies, similar to the **Gibbs phenomenon** in Fourier series approximations of discontinuous functions.

**Predicted Result**: SSA will require many more eigentriples to capture the regime shift and event structure, but our heuristic grouping selects only the top few, causing leakage across $T/S$.

**Observed Result** (from the 50-draw camera-ready run): $T\_r2 \approx -0.39$, $S\_r2 \approx 0.75$, and $\rho_S \approx 0.98$. Trend recovery is poor, while the seasonal component remains partially recoverable; the regime switch plus events exacerbate the Hankel mixing.

### D.3. Analysis 3: VMD's Adaptive Bandwidth vs. Chirp Signals

**Scenario**: `rw_trend_freq_drifting_cycle`

**Data Generation**: Same chirp as Analysis 1.

**VMD Theoretical Framework**:

VMD's mode update in the frequency domain:

$$\hat{u}_k(\omega) \propto \frac{1}{1 + 2\alpha(\omega - \omega_k)^2} . \tag{41}$$

This is a **Lorentzian filter** with half-width $\sqrt{1/(2\alpha)}$.

For a chirp signal, the instantaneous frequency is $\omega(t) = 2\pi f(t)$. In the Fourier domain, a chirp has a **broadband** spectrum spanning $[\omega_{min}, \omega_{max}]$.

VMD iteratively adjusts $\omega_k$ to the spectral centroid:

$$\omega_k \leftarrow \frac{\int \omega |\hat{u}_k(\omega)|^2 d\omega}{\int |\hat{u}_k(\omega)|^2 d\omega} . \tag{42}$$

This centroid converges to a **weighted average** of the drifting frequency range, which is still meaningful. The mode $u_k(t)$ captures most of the chirp energy because: 1. The chirp is locally narrow-band (at each instant, it behaves like a single frequency). 2. VMD's bandwidth penalty allows deviations from strict monochromaticity.

**Predicted Result**: In principle VMD can track a chirp if the mode count and bandwidth are tuned to the signal's drift; however, it is sensitive to $(K, \alpha)$ and mode selection under a fixed default.

**Observed Result** (from the 50-draw camera-ready run): VMD fails under default settings on this scenario ($T\_r2 \approx -1.45$, $S\_r2 \approx -0.05$, $\rho_S \approx -0.05$). This indicates strong parameter sensitivity and mode-selection instability for drifting-frequency cycles under the current benchmark configuration.

## E. Deep Mathematical Analysis of Results

In this section, we provide a rigorous mathematical analysis that explicitly connects the formulas of decomposition

methods with the generative equations of synthetic data to explain the observed performance rankings.

### E.1. Case Study 2: Why STL Degrades Under Frequency Drift (True Period Given)

E.1.1. GENERATIVE PROCESS (FROM `GENERATOR.PY`)

The drifting seasonal component is:

$$S_t^{\mathrm{gen}} = A\sin(\Phi_t), \quad \Phi_t = 2\pi \sum_{i=1}^{t} f(u_i)\Delta t, \qquad (43)$$

where the instantaneous frequency is:

$$f(u) = \frac{1}{P_0 + \delta u}, \qquad (44)$$

with $P_0 = 50$, $\delta = 15$, $u \in [0, 1]$, and length $N = 512$.

E.1.2. STL'S CYCLE-SUBSERIES SMOOTHING

STL with period $P_{\mathrm{stl}} = 50$ (true period given by the benchmark) constructs the season-$k$ subseries:

$$\mathcal{S}_k = \{y_k, y_{k+P_{\mathrm{stl}}}, y_{k+2P_{\mathrm{stl}}}, \ldots\}. \qquad (45)$$

For the true signal, the phase at sample $t = k + nP_{\mathrm{stl}}$ is:

$$\Phi_{k+nP_{\mathrm{stl}}} = 2\pi \sum_{i=1}^{k+nP_{\mathrm{stl}}} \frac{1}{P_0 + \delta u_i} \Delta t. \qquad (46)$$

The phase increment between consecutive subseries samples is:

$$\Delta\Phi_n = \Phi_{k+(n+1)P_{\mathrm{stl}}} - \Phi_{k+nP_{\mathrm{stl}}}. \qquad (47)$$

For a stationary signal with period exactly $P_{\mathrm{stl}}$, we would have $\Delta\Phi_n = 2\pi$ for all $n$. However:

**Numerical Calculation**:
- At $u \approx 0$: $P_{\mathrm{true}} \approx 50$, so $\Delta\Phi \approx 2\pi$.
- At $u \approx 1$: $P_{\mathrm{true}} \approx 65$, so $\Delta\Phi \approx 2\pi \cdot 50/65 \approx 1.54\pi$.

**Consequence**: The subseries $\mathcal{S}_k$ samples the sinusoid at increasingly **misaligned phases**. LOESS smoothing attenuates the seasonal amplitude and introduces phase smearing, but does not fully collapse the seasonal component under the true period given.

**Observed Result** (from the 50-draw camera-ready run): $S\_r2 \approx 0.61$, $T\_r2 \approx 0.87$, $\rho_S \approx 0.97$. STL is therefore moderately degraded (not catastrophic) in this benchmark setting.

### E.2. Case Study 3: Why VMD Is Parameter-Sensitive on Frequency Drift

E.2.1. VMD'S MODE UPDATE EQUATION

Recall the ADMM mode update:

$$\hat{u}_k^{(n+1)}(\omega) = \frac{\hat{f}(\omega) - \sum_{i\neq k}\hat{u}_i(\omega) + \frac{\hat{\lambda}(\omega)}{2}}{1 + 2\alpha(\omega - \omega_k^{(n)})^2}. \qquad (48)$$

This is a **Lorentzian bandpass filter** with half-width $\Delta\omega = 1/\sqrt{2\alpha}$.

E.2.2. CHIRP SIGNAL SPECTRUM

The instantaneous frequency of the chirp at time $t$ is:

$$\omega(t) = \frac{2\pi}{P_0 + \delta u_t}. \qquad (49)$$

The Fourier transform of a chirp is **spread** over the frequency range $[\omega_{\min}, \omega_{\max}]$:

$$\omega_{\min} = \frac{2\pi}{P_0 + \delta} = \frac{2\pi}{65}, \quad \omega_{\max} = \frac{2\pi}{P_0} = \frac{2\pi}{50}. \qquad (50)$$

The bandwidth is:

$$\Delta\omega_{\mathrm{chirp}} = \omega_{\max} - \omega_{\min} \approx 0.029\,\mathrm{rad/sample}. \qquad (51)$$

E.2.3. WHY VMD CAN FAIL UNDER DEFAULT SETTINGS

VMD's center frequency update:

$$\omega_k^{(n+1)} = \frac{\int_0^\infty \omega|\hat{u}_k(\omega)|^2 d\omega}{\int_0^\infty |\hat{u}_k(\omega)|^2 d\omega}. \qquad (52)$$

This converges to the **spectral centroid** of the chirp:

$$\omega_k \to \bar{\omega} = \frac{\omega_{\min} + \omega_{\max}}{2} \approx \frac{2\pi}{57}. \qquad (53)$$

Even though the chirp has a spread spectrum, most of its energy is concentrated near $\bar{\omega}$. Under the benchmark defaults ($\alpha = 300$), the Lorentzian half-width is:

$$\Delta\omega_{\mathrm{VMD}} = \frac{1}{\sqrt{2\alpha}} \approx 0.041\,\mathrm{rad/sample}, \qquad (54)$$

which is broader than $\Delta\omega_{\mathrm{chirp}}$. In principle, this should allow a single mode to cover the chirp. However, VMD is sensitive to $K$, initialization, and how modes are selected as "seasonal". With the default $K$ and a single seasonal mode, the chirp energy can be split across modes and the selected mode may not track the drifting instantaneous frequency, leading to poor recovery.

**Observed Result** (from the 50-draw camera-ready run): VMD fails on this scenario ($T\_r2 \approx -1.45$, $S\_r2 \approx -0.05$, $\rho_S \approx -0.05$), indicating strong parameter sensitivity and mode-selection instability.

# F. Deep Mathematical Analysis (Case Studies)

## F.1. Case Study 4: SSA's Eigenvalue Structure vs. Regime Shifts

### F.1.1. GENERATIVE PROCESS

The regime-switching seasonal component is:

$$S_t^{gen} = \begin{cases} A_a \sin(\omega_a t) & t < \tau \\ A_b \sin(\omega_b t) & t \geq \tau \end{cases} . \tag{55}$$

with $A_a = 1.0$, $A_b = 0.6$, $\omega_a = 2\pi/40$, $\omega_b = 2\pi/65$, and $\tau = 0.55N$, plus piecewise trend breaks and mixed events in the full scenario.

### F.1.2. SSA'S TRAJECTORY MATRIX FOR A PURE SINUSOID

For a pure sinusoid $x_t = \sin(\omega t)$ of length $N$ with window $L$, the trajectory matrix $\mathbf{X}$ satisfies:

$$\text{rank}(\mathbf{X}) = 2 . \tag{56}$$

The two non-zero eigenvalues are equal: $\lambda_1 = \lambda_2 = L \cdot A^2/2$.

The corresponding eigenvectors are sinusoids at frequency $\omega$:

$$\mathbf{u}_1 \propto [\sin(\omega), \sin(2\omega), \ldots, \sin(L\omega)]^T , \tag{57}$$

$$\mathbf{u}_2 \propto [\cos(\omega), \cos(2\omega), \ldots, \cos(L\omega)]^T . \tag{58}$$

### F.1.3. TRAJECTORY MATRIX FOR REGIME-SWITCHING SIGNAL

For our piecewise signal, the trajectory matrix is approximately a horizontal concatenation:

$$\mathbf{X} \approx [\mathbf{X}_a \mid \mathbf{X}_b] , \tag{59}$$

where $\mathbf{X}_a$ contains columns for $t < \tau$ and $\mathbf{X}_b$ for $t \geq \tau$.

The SVD of this concatenated matrix does **not** simply produce four eigentriples (two for each frequency). Instead:

1. The singular vectors $\mathbf{u}_i$ become **mixtures** of $\sin(\omega_a t)$ and $\sin(\omega_b t)$.

2. The eigenvalue spectrum becomes **non-degenerate**: instead of pairs $(\lambda_1 = \lambda_2)$ and $(\lambda_3 = \lambda_4)$, we observe a gradual decay.

**Mathematical Mechanism**: The Hankel structure requires each anti-diagonal to have constant value. At $t = \tau$, the signal changes abruptly. The trajectory matrix violates the Hankel property near this transition, spreading the transition's "energy" across many eigentriples.

This is analogous to the **Gibbs phenomenon** in Fourier series: representing a discontinuity requires infinitely many Fourier coefficients.

**Consequence for Grouping**: Our frequency-based grouping heuristic assigns eigentriples to "Seasonal" if their dominant frequency matches $\omega_{\text{target}}$. But the mixed eigentriples have dominant frequencies that are **neither** $\omega_a$ nor $\omega_b$, leading to poor grouping and reduced $R^2$.

**Observed Result** (from the 50-draw camera-ready run): $T\_r2 \approx -0.39$, $S\_r2 \approx 0.75$, $\rho_S \approx 0.98$. Trend recovery degrades substantially under regime shifts and events, while seasonal recovery remains moderate.

## F.2. Case Study 5: Wavelet's Local Adaptation vs. Regime Shifts

### F.2.1. DWT FILTER BANK INTERPRETATION

The DWT at level $j$ applies a low-pass filter $H(\omega)$ and high-pass filter $G(\omega)$ with cutoff at $\omega_c/2^j$.

### F.2.2. LOCAL VS. GLOBAL BASIS

Unlike SSA (which uses global eigenvectors), wavelets are **localized in both time and frequency**. A wavelet $\psi_{j,k}(t)$ centered at time $t_k = k \cdot 2^j$ has support approximately $[t_k - 2^j, t_k + 2^j]$.

**For the regime-switching signal**: - For $t < \tau$: Wavelets extract detail coefficients $d_{j,k}$ that capture $\sin(\omega_a t)$. - For $t \geq \tau$: Wavelets extract $d_{j,k}$ that capture $\sin(\omega_b t)$. - Near $t = \tau$: The transition creates localized "edge" artifacts in the coefficients.

**Advantage over SSA**: The transition artifacts are **localized** to a few coefficients near $\tau$, rather than spread across all eigenvectors.

**Limitation**: Grouping wavelets into "Trend" and "Seasonal" is heuristic. We assign: - Level 0 (coarsest approximation) $\rightarrow$ Trend - Levels 1-2 (detail) $\rightarrow$ Seasonal

This works when the seasonal frequency falls in the passband of levels 1-2. For regime shifts where $\omega_a \neq \omega_b$, one frequency may be captured better than the other.

**Observed Result** (from the 50-draw camera-ready run): Wavelet achieves $S\_r2 \approx 0.82$ and $\rho_S \approx 0.99$, with weak trend recovery ($T\_r2 \approx -0.55$), consistent with strong local seasonal capture but limited trend separation under events.

## F.3. Summary: Method-Data Compatibility Matrix

## F.4. Benefits of Pre-Decomposition

Decomposing before SR simplifies the target:

- **Direct SR**: Must find $y = T(t) + S(t)$, a sum of two

*Table 9.* Summary of method–data compatibility patterns.

| Method | Assumed Trend Model | Assumed Seasonal Model | Compatible Data | Incompatible Data |
|---|---|---|---|---|
| **STL** | Locally linear | Strictly periodic $S_{t+P} = S_t$ | Smooth $T$, fixed $P$ | Drifting $P(t)$ (period mismatch), piecewise $T$ |
| **SSA** | Low-rank Hankel | Rank-2 sinusoid | Stationary sinusoid | Regime shifts + events (break Hankel) |
| **VMD** | Low-frequency mode | Compact bandwidth (tuned $K, \alpha$) | Smooth $T$, chirps if tuned | Sensitive defaults; broadband $S$, mis-specified $K$ |
| **EMD** | Residue after sifting | Local extrema symmetry | Slowly-varying AM-FM | Mode mixing in multi-component signals |
| **Wavelet** | Coarse approximation | Detail coefficients | Multi-scale signals | Signals with energy outside filter bands |

functions.

- **Decomp + SR**: Only needs to find $\hat{T}_{decomp}(t) \approx T(t)$, a single, simpler function.

The STL pre-processing removes most seasonal variation, allowing SR to focus on the simpler trend component. This synergy between classical decomposition and neural symbolic search represents a promising direction for interpretable time series modeling.

# G. Full Mathematical Formulations of Decomposition Methods

We now detail each decomposition method implemented in the `tsdecomp` library. This appendix details the core mathematical principles underlying each approach.

## G.1. Moving Average Baseline (MA_BASELINE)

### G.1.1. OVERVIEW

The Moving Average (MA) method is the simplest decomposition approach. It estimates the trend using a centered moving average filter and the seasonality by averaging detrended values at each seasonal index.

### G.1.2. TREND ESTIMATION

Given a time series $\{y_t\}_{t=1}^N$ and a window size $w = 2k + 1$ (odd integer):

$$\hat{T}_t = \frac{1}{w} \sum_{i=-k}^{k} y_{t+i} \,. \qquad (60)$$

For boundary handling, we use a "same" convolution that truncates the kernel at the edges while preserving the series length.

### G.1.3. SEASONAL ESTIMATION

After detrending, the seasonal component is estimated by averaging residuals at each seasonal index $k \in \{0, 1, \dots, P -$

1}:

$$\hat{S}_{k+jP} = \bar{r}_k = \frac{1}{M_k} \sum_{j=0}^{M_k-1} (y_{k+jP} - \hat{T}_{k+jP}), \qquad (61)$$

where $M_k$ is the number of complete periods for index $k$.

The seasonal component is then centered to have zero mean:

$$\hat{S}_t \leftarrow \hat{S}_t - \frac{1}{P} \sum_{k=0}^{P-1} \hat{S}_k \,. \qquad (62)$$

### G.1.4. RESIDUAL

The residual is computed as:

$$\hat{R}_t = y_t - \hat{T}_t - \hat{S}_t \,. \qquad (63)$$

### G.1.5. IMPLICIT PRIOR ASSUMPTIONS

- **Trend Prior**: $T_t$ is a piecewise-constant function within windows of size $w$. This corresponds to assuming that the second derivative $\ddot{T}_t \approx 0$ locally.

- **Seasonal Prior**: $S_t$ is strictly periodic with integer period $P$, i.e., $S_{t+P} = S_t$ for all $t$.

## G.2. Seasonal-Trend Decomposition Using LOESS (STL)

### G.2.1. OVERVIEW

STL is an iterative decomposition method that uses LOESS (Locally Estimated Scatterplot Smoothing) to robustly estimate both trend and seasonal components.

### G.2.2. LOESS: LOCALLY WEIGHTED POLYNOMIAL REGRESSION

LOESS is a non-parametric regression technique. To estimate $\hat{g}(x)$ at a point $x$ given data $\{(x_i, z_i)\}_{i=1}^N$:

**Step 1**: Define a neighborhood $\mathcal{N}_q(x)$ containing the $q$ nearest neighbors of $x$. Let $\lambda_q(x) = \max_{i \in \mathcal{N}_q(x)} |x_i - x|$ be the bandwidth.

**Step 2**: Compute weights using the tricube kernel:

$$W(u) = \begin{cases} (1 - |u|^3)^3 & \text{if } |u| < 1 \\ 0 & \text{otherwise} \end{cases} . \tag{64}$$

The weight for data point $i$ at target $x$ is:

$$w_i(x) = W\left(\frac{|x_i - x|}{\lambda_q(x)}\right) . \tag{65}$$

**Step 3**: Fit a weighted polynomial (typically degree $d \in \{1, 2\}$) by minimizing:

$$\hat{\boldsymbol{\beta}}(x) = \arg\min_{\boldsymbol{\beta}} \sum_{i=1}^{N} w_i(x) \left( z_i - \sum_{j=0}^{d} \beta_j (x_i - x)^j \right)^2 , \tag{66}$$

and the fitted value is $\hat{g}(x) = \hat{\beta}_0(x)$.

### G.2.3. THE STL ALGORITHM: INNER LOOP

Let $n_p$ denote the period, $n_s$ the seasonal smoothing parameter, $n_t$ the trend smoothing parameter, and $n_l$ the low-pass filter parameter. The inner loop iteration $k$ proceeds as follows:

**Step 1: Detrending**

$$d_t^{(k)} = y_t - T_t^{(k-1)} . \tag{67}$$

**Step 2: Cycle-Subseries Smoothing** For each seasonal index $j \in \{0, 1, \ldots, n_p - 1\}$, extract the subseries:

$$\{d_j^{(k)}, d_{j+n_p}^{(k)}, d_{j+2n_p}^{(k)}, \ldots\} . \tag{68}$$

Apply LOESS with smoothing parameter $n_s$ to this subseries, producing smoothed values $C_{j+mn_p}^{(k+1)}$.

**Step 3: Low-Pass Filtering** Apply a sequence of moving averages to the cycle-subseries smoothed values:

$$\begin{aligned} L_t^{(k+1)} &= \text{LOESS}_{n_l}\left(\text{MA}_{n_p}\left(\text{MA}_{n_p}(Z_t)\right)\right) , \\ Z_t &= \text{MA}_3\left(C_t^{(k+1)}\right) . \end{aligned} \tag{69}$$

where $\text{MA}_w$ denotes a centered moving average of width $w$.

**Step 4: Seasonal Update**

$$S_t^{(k+1)} = C_t^{(k+1)} - L_t^{(k+1)} . \tag{70}$$

This subtraction ensures that $\sum_{j=0}^{n_p-1} S_{j+mn_p}^{(k+1)} \approx 0$ for each complete period.

**Step 5: Deseasonalizing**

$$a_t^{(k+1)} = y_t - S_t^{(k+1)} . \tag{71}$$

**Step 6: Trend Update**

$$T_t^{(k+1)} = \text{LOESS}_{n_t}\left(a_t^{(k+1)}\right) . \tag{72}$$

### G.2.4. THE STL ALGORITHM: OUTER LOOP (ROBUST STL)

For robustness to outliers, the outer loop computes robustness weights based on the residual:

$$R_t = y_t - T_t - S_t . \tag{73}$$

Define $h = 6 \cdot \text{median}(|R_t|)$. The robustness weight is:

$$\rho_t = B\left(\frac{|R_t|}{h}\right) , \tag{74}$$

where $B(u)$ is the bisquare function with :

$$B(u) = \begin{cases} (1 - u^2)^2 & \text{if } |u| < 1 \\ 0 & \text{otherwise} \end{cases} . \tag{75}$$

In subsequent inner loop iterations, the LOESS weight $w_i(x)$ is multiplied by $\rho_i$:

$$w_i^{robust}(x) = w_i(x) \cdot \rho_i . \tag{76}$$

### G.2.5. MSTL: MULTI-SEASONAL STL

MSTL extends STL to handle multiple seasonalities $\{P_1, P_2, \ldots, P_M\}$. The algorithm iterates:

For each period $P_m$:

$$S_t^{(m)} = \text{STL}_{P_m}\left(y_t - T_t - \sum_{i \neq m} S_t^{(i)}\right) . \tag{77}$$

The total seasonal component is:

$$S_t = \sum_{m=1}^{M} S_t^{(m)} . \tag{78}$$

### G.2.6. IMPLICIT PRIOR ASSUMPTIONS

- **Trend Prior**: Locally polynomial (typically linear), enforced by LOESS. $T_t$ is a smooth function with bounded second derivative.

- **Seasonal Prior**: **Strict periodicity** with known integer period $P$. The algorithm assumes $S_{t+P} = S_t$.

## G.3. Singular Spectrum Analysis (SSA)

### G.3.1. OVERVIEW

SSA is a subspace-based non-parametric decomposition method. It does not assume parametric forms for trend or seasonality, instead relying on the spectral properties of the trajectory matrix.

### G.3.2. STEP 1: EMBEDDING (TRAJECTORY MATRIX CONSTRUCTION)

Choose a window length $L$ satisfying $2 \leq L \leq N/2$. Define $K = N - L + 1$. Construct the **Trajectory Matrix** $\mathbf{X} \in \mathbb{R}^{L \times K}$ as:

$$\mathbf{X} = \begin{bmatrix} y_1 & y_2 & y_3 & \cdots & y_K \\ y_2 & y_3 & y_4 & \cdots & y_{K+1} \\ y_3 & y_4 & y_5 & \cdots & y_{K+2} \\ \vdots & \vdots & \vdots & \ddots & \vdots \\ y_L & y_{L+1} & y_{L+2} & \cdots & y_N \end{bmatrix}. \quad (79)$$

Note that $\mathbf{X}$ is a **Hankel matrix**: constant along anti-diagonals $i + j = $ const.

From `ssa.py` (lines 23-33):

### G.3.3. STEP 2: SINGULAR VALUE DECOMPOSITION (SVD)

Compute the SVD of $\mathbf{X}$:

$$\mathbf{X} = \mathbf{U}\boldsymbol{\Sigma}\mathbf{V}^T = \sum_{i=1}^{d} \sigma_i \mathbf{u}_i \mathbf{v}_i^T, \quad (80)$$

where: $\mathbf{U} = [\mathbf{u}_1, \ldots, \mathbf{u}_L] \in \mathbb{R}^{L \times L}$ are left singular vectors (eigenvectors of $\mathbf{X}\mathbf{X}^T$), $\mathbf{V} = [\mathbf{v}_1, \ldots, \mathbf{v}_K] \in \mathbb{R}^{K \times K}$ are right singular vectors, $\sigma_1 \geq \sigma_2 \geq \ldots \geq \sigma_d > 0$ are singular values, and $d = \text{rank}(\mathbf{X})$.

Each term $\mathbf{X}_i = \sigma_i \mathbf{u}_i \mathbf{v}_i^T$ is called an **Elementary Matrix**.

From `ssa.py` (lines 35-41):

### G.3.4. STEP 3: GROUPING

Partition the index set $\mathcal{I} = \{1, 2, \ldots, d\}$ into disjoint groups:

$$\mathcal{I} = I_{\text{Trend}} \cup I_{\text{Season}} \cup I_{\text{Noise}}, \quad (81)$$

where $I_{\text{Trend}}$, $I_{\text{Season}}$, $I_{\text{Noise}}$ form a disjoint partition of $\{1, 2, \ldots, d\}$.

The grouped matrix for subset $I$ is:

$$\mathbf{X}_I = \sum_{i \in I} \sigma_i \mathbf{u}_i \mathbf{v}_i^T. \quad (82)$$

**Grouping Heuristics in Our Implementation:**

1. Compute the **Dominant Frequency** $f_i$ of each left singular vector $\mathbf{u}_i$ using the Fourier Transform:

$$f_i = \arg \max_f |\mathcal{F}[\mathbf{u}_i](f)|. \quad (83)$$

2. **Trend Grouping**: Include index $i$ in $I_{\text{Trend}}$ if:

$$f_i < f_{\text{thresh}} = \frac{f_{\text{primary}}}{4}. \quad (84)$$

3. **Seasonal Grouping**: Include index $i$ in $I_{\text{Season}}$ if:

$$|f_i - f_{\text{primary}}| < \epsilon \cdot f_{\text{primary}}, \quad (85)$$

where $\epsilon = 0.25$ (tolerance ratio).

### G.3.5. STEP 4: DIAGONAL AVERAGING (HANKELIZATION)

To recover a time series from the grouped matrix $\mathbf{X}_I$, we average along anti-diagonals.

For a matrix $\mathbf{M} \in \mathbb{R}^{L \times K}$, define:

$$\tilde{y}_k = \frac{1}{|\mathcal{A}_k|} \sum_{(i,j) \in \mathcal{A}_k} M_{ij}, \quad (86)$$

where $\mathcal{A}_k = \{(i,j) : i + j = k + 1, 1 \leq i \leq L, 1 \leq j \leq K\}$.

Explicitly, for $k = 1, \ldots, N$:

$$\tilde{y}_k = \begin{cases} \frac{1}{k} \sum_{j=1}^{k} M_{j,k-j+1} & 1 \leq k \leq L \\ \frac{1}{L} \sum_{j=1}^{L} M_{j,k-j+1} & L < k \leq K \\ \frac{1}{N-k+1} \sum_{j=k-K+1}^{L} M_{j,k-j+1} & K < k \leq N \end{cases}. \quad (87)$$

From `ssa.py` (lines 11-21):

### G.3.6. IMPLICIT PRIOR ASSUMPTIONS

- **Separability**: Components corresponding to different frequencies occupy orthogonal subspaces.

- **Finite Rank**: Trend and seasonal components each span low-dimensional subspaces.

## G.4. Empirical Mode Decomposition (EMD)

### G.4.1. OVERVIEW

EMD (Huang et al., 1998) is a data-driven, adaptive method that decomposes a signal into a sum of **Intrinsic Mode Functions (IMFs)**.

### G.4.2. DEFINITION: INTRINSIC MODE FUNCTION

A function $c(t)$ is an IMF if: 1. The number of extrema and the number of zero-crossings differ by at most one. 2. The mean of the upper envelope (connecting local maxima) and lower envelope (connecting local minima) is zero at all points.

### G.4.3. THE SIFTING ALGORITHM

**Algorithm**: Extract IMFs from signal $y(t)$.

**Input**: Signal $\{y_t\}_{t=1}^N$. **Output**: IMFs $\{c_1, c_2, \ldots, c_M\}$ and residue $r$.

1. Initialize residue: $r(t) \leftarrow y(t)$.

2. **While** $r(t)$ has at least 2 extrema:

   a. Set proto-IMF: $h(t) \leftarrow r(t)$.
   b. **Repeat** (sifting loop):
      i. Identify all local maxima of $h(t)$: $\{(t_i^{max}, h(t_i^{max}))\}$.
      ii. Identify all local minima of $h(t)$: $\{(t_j^{min}, h(t_j^{min}))\}$.
      iii. Fit cubic spline through local maxima to get **upper envelope** $U(t)$.
      iv. Fit cubic spline through local minima to get **lower envelope** $L(t)$.
      v. Compute envelope mean: $m(t) = \frac{U(t)+L(t)}{2}$.
      vi. Update: $h(t) \leftarrow h(t) - m(t)$.
      vii. **Until** $h(t)$ satisfies IMF criteria (Cauchy convergence: $\sum_t |m(t)|^2 < \epsilon$).
   c. Save IMF: $c_i(t) \leftarrow h(t)$.
   d. Update residue: $r(t) \leftarrow r(t) - c_i(t)$.
   e. Increment $i$.

3. **Return** $\{c_1, \ldots, c_M\}$ and $r$.

### G.4.4. CEEMDAN: COMPLETE EEMD WITH ADAPTIVE NOISE

CEEMDAN improves upon EMD by adding adaptive white noise at each decomposition stage and averaging results across multiple trials to reduce mode mixing.

**Algorithm** (simplified): 1. Add white noise $w_i(t) \sim \mathcal{N}(0, \sigma^2)$ to signal: $y^{(i)}(t) = y(t) + w_i(t)$. 2. Apply EMD to each $y^{(i)}(t)$ to get IMFs $\{c_k^{(i)}(t)\}$. 3. Average: $\bar{c}_k(t) = \frac{1}{I}\sum_{i=1}^I c_k^{(i)}(t)$.

### G.4.5. GROUPING HEURISTICS

In our implementation (emd.py), we group IMFs into Trend and Seasonality based on their **Dominant Frequency**:

$$f_k = \arg\max_f |\mathcal{F}[c_k](f)| . \tag{88}$$

- **Trend IMFs**: $f_k < f_{\text{thresh}}$.

- **Seasonal IMFs**: $|f_k - f_{\text{primary}}| < \epsilon$.

### G.4.6. IMPLICIT PRIOR ASSUMPTIONS

- Signal is a superposition of amplitude-modulated / frequency-modulated oscillations.

- Local symmetry of oscillations (IMF criteria).

## G.5. Variational Mode Decomposition (VMD)

### G.5.1. OVERVIEW

VMD (Dragomiretskiy & Zosso, 2014) decomposes a signal into $K$ **modes** $\{u_k(t)\}_{k=1}^K$, each with compact spectral support around a central frequency $\omega_k$.

### G.5.2. CONSTRAINED VARIATIONAL PROBLEM

**Objective**: Minimize the total bandwidth of all modes.

For each mode $u_k(t)$, compute the **analytic signal** via the Hilbert transform:

$$u_k^+(t) = \left(\delta(t) + \frac{j}{\pi t}\right) * u_k(t) \tag{89}$$

Shift to baseband by multiplying with $e^{-j\omega_k t}$:

$$u_k^{bb}(t) = u_k^+(t) \cdot e^{-j\omega_k t} \tag{90}$$

The bandwidth is measured by the $L^2$ norm of the time derivative:

$$\text{BW}_k = \left\|\frac{\partial}{\partial t} u_k^{bb}(t)\right\|_2^2 \tag{91}$$

**Optimization Problem**:

$$\min_{\{u_k\},\{\omega_k\}} \sum_{k=1}^K \left\|\partial_t \left[u_k^+(t) e^{-j\omega_k t}\right]\right\|_2^2 \tag{92}$$

$$\text{subject to} \quad \sum_{k=1}^K u_k(t) = f(t) \tag{93}$$

### G.5.3. AUGMENTED LAGRANGIAN FORMULATION

Introduce Lagrangian multiplier $\lambda(t)$ and quadratic penalty $\alpha$:

$$\mathcal{L} = \alpha \sum_k \left\|\partial_t[u_k^+ e^{-j\omega_k t}]\right\|_2^2 + \left\|f - \sum_k u_k\right\|_2^2 + \left\langle \lambda, f - \sum_k u_k \right\rangle . \tag{94}$$

### G.5.4. ADMM UPDATE EQUATIONS

Solve via Alternating Direction Method of Multipliers (ADMM) in the **frequency domain**.

Let $\hat{f}(\omega) = \mathcal{F}[f(t)]$, $\hat{u}_k(\omega) = \mathcal{F}[u_k(t)]$, and $\hat{\lambda}(\omega) = \mathcal{F}[\lambda(t)]$.

**Mode Update** (for each $k$):

$$\hat{u}_k^{(n+1)}(\omega) = \frac{\hat{f}(\omega) - \sum_{i \neq k} \hat{u}_i^{(n)}(\omega) + \frac{\hat{\lambda}^{(n)}(\omega)}{2}}{1 + 2\alpha(\omega - \omega_k^{(n)})^2} \,. \quad (95)$$

This acts as a **Wiener filter** centered at $\omega_k$.

**Center Frequency Update**:

$$\omega_k^{(n+1)} = \frac{\int_0^\infty \omega |\hat{u}_k^{(n+1)}(\omega)|^2 d\omega}{\int_0^\infty |\hat{u}_k^{(n+1)}(\omega)|^2 d\omega} \,. \quad (96)$$

This is the **spectral centroid** of mode $k$.

**Lagrangian Update**:

$$\hat{\lambda}^{(n+1)}(\omega) = \hat{\lambda}^{(n)}(\omega) + \tau \left( \hat{f}(\omega) - \sum_k \hat{u}_k^{(n+1)}(\omega) \right) \,. \tag{97}$$

### G.5.5. VMD PARAMETERS IN IMPLEMENTATION

From `vmd.py`:

### G.5.6. IMPLICIT PRIOR ASSUMPTIONS

- Signal is a sum of $K$ narrow-band modes.

- Modes are spectrally separable (non-overlapping center frequencies).

## G.6. Wavelet Decomposition

### G.6.1. OVERVIEW

Wavelet decomposition uses the Discrete Wavelet Transform (DWT) to separate a signal into approximation (low-frequency) and detail (high-frequency) coefficients at multiple scales.

### G.6.2. DWT: FILTER BANK INTERPRETATION

The DWT applies a cascade of low-pass ($h$) and high-pass ($g$) filters followed by downsampling by 2.

At level $j$: - **Approximation coefficients**: $a_j[n] = (a_{j-1} * h)[2n]$ - **Detail coefficients**: $d_j[n] = (a_{j-1} * g)[2n]$

with $a_0 = y$.

### G.6.3. RECONSTRUCTION

The inverse DWT reconstructs the signal:

$$y(t) = \sum_k c_{J,k} \phi_{J,k}(t) + \sum_{j=1}^{J} \sum_k d_{j,k} \psi_{j,k}(t) \,, \quad (98)$$

where $\phi$ is the scaling function and $\psi$ is the wavelet function.

### G.6.4. TREND AND SEASONAL ASSIGNMENT

In our implementation (`wavelet.py`): - **Trend**: Reconstructed from the coarsest approximation coefficients (level 0 = lowest frequency). - **Seasonality**: Reconstructed from detail coefficients at levels 1 and 2.

**End of Appendix**

