# OpenReview forum: "Time-Series Decomposition as a Standalone Task: A Mechanism-Driven Diagnostic Benchmark"
_ICML.cc/2026/Conference — ICML 2026 regular_

### Official Review · Reviewer_FFgj · 2026-03-09

**Soundness:** 3
**Presentation:** 3
**Significance:** 3
**Originality:** 4
**Overall Recommendation:** 5
**Confidence:** 4

**Summary:**

This paper reframes time-series decomposition as a rigorous, standalone machine learning task rather than an informal preprocessing step. The authors present a novel benchmark utilizing synthetic data generation, which allows for the evaluation of methods against known ground-truth components, including trend, seasonality, and residuals. They systematically assess six families of classical decomposition methods across both stationary and non-stationary regimes, such as frequency drift and regime switching. Furthermore, the authors demonstrate through a downstream symbolic regression task that accurate decomposition is critical for uncovering physical laws within data.

**Compliance With Llm Reviewing Policy:**

Affirmed.

**Final Justification:**

The paper originally presented a highly original premise evaluating time-series decomposition as a rigorous, standalone task, but my initial rating was held back by the lack of neural baselines, the reliance on purely synthetic data, and the strict use of default hyperparameters.

The authors provided an exceptional, experiment-driven rebuttal that thoroughly resolved every single one of my concerns. Most impressively, they extracted and evaluated neural decomposition blocks from over a dozen recent architectures, yielding the highly significant insight that these learned modules largely mirror classical biases. Furthermore, the new semi-synthetic track (with great practical insights into CEEMDAN's mode-mixing), the bounded tuning budgets, and the period-misspecification ablations drastically improve the paper's soundness and real-world relevance.

The authors put in the hard work to patch every experimental blind spot. This is now a robust, highly valuable benchmark for the time-series community, and I confidently recommend acceptance.

**Key Questions For Authors:**

1. Are there any plans to include neural baselines (like the series decomposition block from Autoformer) to see if learned priors overcome the limitations of classical fixed-period or global-basis assumptions?
2. What happens to VMD's performance if you give it even a minimal hyperparameter sweep (e.g., a small grid search over K and α )?
3. Have you considered running a semi-synthetic evaluation to prove that your diagnostic insights transfer to real-world noise and structures?
4. How rapidly does the performance of STL/MSTL degrade if the injected period is slightly misspecified or noisy?

**Limitations:**

yes

**Strengths And Weaknesses:**

**Strengths**
* The authors accurately observe that decomposition is frequently evaluated through visual inspection of output plausibility. By formalizing this process as a component-recovery task with ground-truth targets, the paper makes a significant contribution to the field.
* Employing a combination of complementary metrics is highly effective. By evaluating scale (R2), shape (DTW), and phase or frequency robustness (max-lag and spectral correlations), the benchmark provides diagnostic insight into algorithmic failure modes rather than offering only binary pass or fail outcomes.
* Rather than merely ranking the methods, the paper effectively maps the inductive biases of the algorithms to specific failure regimes. For example, the clear demonstration of STL's breakdown under frequency drift, attributable to its fixed-period assumption, is particularly valuable for practitioners.
* Appendices C, D, and E demonstrate a strong understanding of the mathematical mechanics underlying the tested algorithms. The authors leverage this expertise to provide clear explanations for the empirical behavioThe CO₂ experiment serves as a compelling demonstration. Illustrating that direct symbolic regression fails to identify the seasonal law, while a decompose-then-regress pipeline successfully recovers the sin(2πt) function, provides strong evidence for the downstream value of this approach. downstream value of this work.



**Weaknesses**

* The time-series community has spent the last few years building neural architectures that rely heavily on learned decomposition blocks (e.g., Autoformer, FEDformer, TimesNet). Even if these are technically built for forecasting, extracting their decomposition outputs and benchmarking them against classical methods like STL or SSA is necessary for this paper to reflect the current ML landscape.
* The core benchmark does not incorporate real data. Real-world datasets, such as CO₂ and tides, are only utilized in the symbolic regression demonstration. Consequently, it is difficult to fully trust the method selection guidance, as it has not been evaluated on real data where noise distributions are complex and ground-truth components are only approximate. Incorporating a semi-synthetic track, for example by superimposing a known synthetic seasonal component onto a real-world trend, would significantly strengthen the evaluation.
* Evaluating everything "out of the box" makes some of the comparisons fundamentally unfair. For example, the paper reports a catastrophic failure for VMD on non-stationary regimes (Trend R2 around -1.99). However, as the authors admit in the appendix, this happens mostly because the default bandwidth parameter (α) is too narrow for a chirp signal. Punishing an algorithm because its default config doesn't perfectly match the synthetic generator risks misleading people reading the benchmark.
* The benchmark feeds the exact, true base period to methods that require it (like STL). In the real world, period estimation and decomposition are deeply entangled. Giving STL the exact period gives it a massive, artificial advantage over fully data-driven methods like EMD or SSA.

---

> ### Author Rebuttal · Authors · 2026-03-30
>
> We thank the reviewer. The original time series decomposition package was already included in the submission attachment. During review, we continued extending the benchmark and software stack, including an accelerated C++-backed fasterVersion, and used it to run the additional experiments reported below. We view this benchmark as infrastructure for studying how models extract core trend and periodic structure inside newer architectures. The rebuttal experiments directly test the main concerns here: transfer, fairness, period information, and modern-baseline coverage.
>
> # 1. Neural block from recent neural time-series models.
> We also want to clarify why the initial submission did not include a large neural-decomposition leaderboard. To the best of our knowledge, there is currently no dedicated deep-learning family whose primary purpose is standalone time-series decomposition and that is benchmarked as such in the same way that STL is decomposition method. If the reviewer is aware of a direct counterexample, we would be happy to discuss it in revision. Recent neural work instead uses decomposition more often as an internal block, module, or representation layer for extracting trend, seasonality, or multiscale structure inside
> larger forecasting models. Under the same protocol, we added the following neural experiments inlcude **Autoformer block** experiments and a backprop-trained learned-prior -interpretable **N-BEATS block**, with explicit trend and seasonality stacks.
> | Item | Result |
> | --- | --- |
> | Tuned synthetic | Autoformer_block > SSA > N-BEATS_block > CEEMDAN > MA > STL > VMD > Wavelet |
> | Real-data caveat | N-BEATS still remains weak on real seasonal recovery: CO$_2$ trend/seasonal $R^2 = 0.9998 / -0.161$ and tides seasonal $R^2 = -0.042$ with target-frequency coverage $0$ and M2/S2 separation $0$ |
>
> So the rebuttal now includes concrete neural methods evidence, but it still does not indicate that learned priors automatically remove the fixed-period or global-basis limitations. **Concretely, the Autoformer block no longer leads on the drifting-frequency scenario, while interpretable N-BEATS fails most clearly on drifting-frequency and single-sine, with strongly negative trend/seasonal R^2 and near-zero seasonal spectral/max-lag agreement.** We will therefore present this as partial but concrete evidence about the current ML landscape, not as a claim that neural coverage is now exhaustive.
>
> # 2. What happens under a minimal hyperparameter sweep for VMD?
>
> **See the rebuttal of Reviewer WfuW 2. Parameter-tuned version**
>
> # 3. Semi-synthetic evaluation and transfer to real-world noise and structures.
>
> We give the fullest synthetic-to-real discussion in our reply to Reviewer zQyK, Q1. The direct answer here is yes: we added a semi-synthetic transfer track that injects known mechanisms into 6 monthly real backgrounds, with 3 mechanisms, 2 background scales, and 8 windows per setting at length 480. To compare like with like, we summarize both the original full synthetic benchmark and this semi-synthetic track by mean within-setting rank on the same recovery metrics available in both tracks:
>
> | Setting | Aggregate order (best -> worst) |
> | --- | --- |
> | Semi-synthetic | SSA > MA > STL > VMD > CEEMDAN > Wavelet|
>
> The most stable result is that SSA remains the top in both settings. The largest drops is CEEMDAN, whereas MA and STL rise relative to pure synthetic, with the same default VMD configuration and the same wavelet setting. We therefore position this as partial transfer evidence rather than a claim that the semi-synthetic track reproduces the exact synthetic leaderboard.
>
> # 4. How rapidly does STL/MSTL degrade under period misspecification?
> On the true-period advantage, we added period-estimation and misspecification analyses. Our main non-oracle setting estimates the period by FFT on the first 60\% of the series, and we additionally sweep explicit misspecification up to $\pm 20\%$. Rather than collapsing this into one score, the table below reports the main recorded recovery metrics:
>
> | Method | Setting | Trend $R^2$ | Sea. spectral | Sea. max-lag |
> | --- | --- | ---: | ---: | ---: |
> | STL | True | $0.954$ | $0.913$ | $0.889$ |
> | STL | FFT | $0.884$ | $0.823$ | $0.798$ |
> | STL | $+20\%$ | $0.928$ | $0.512$ | $0.405$ |
> | MSTL | True | $0.987$ | $0.958$ | $0.913$ |
> | MSTL | FFT | $0.886$ | $0.815$ | $0.790$ |
> | MSTL | $+20\%$ | $0.940$ | $0.467$ | $0.338$ |
>
> The main result is that STL/MSTL remain competitive with reasonable FFT-based period estimates, but their seasonal metrics degrade sharply under larger misspecification, and ACF estimation can fail badly. We will therefore avoid implying that access to the true period is realistic, and instead describe them as performing well when period information is available or estimated reasonably.

---

> > ### Author Rebuttal · Reviewer_FFgj · 2026-04-01
> >
> > I thank the authors for a substantive, experiment-driven rebuttal.
> >
> >
> > **Q4 (period misspecification)** and **Q2 (VMD tuning)** are convincingly addressed. The period degradation table is a practical contribution that belongs in the main text, and the bounded-budget tuning honestly shows VMD improves without changing the overall story.
> >
> > **Q1 (neural baselines)** is partially addressed. Two architectures (Autoformer block, N-BEATS) are a good start, and the finding that learned priors don't automatically overcome classical limitations is informative. I acknowledge the authors' point that no dedicated neural decomposition method currently exists in the same standalone sense as STL. The extracted Autoformer block and N-BEATS results are informative, particularly the finding that learned priors do not automatically overcome classical limitations. This is sufficient for the current submission, though broader coverage in future versions of the benchmark would add value.
> >
> > **Q3 (semi-synthetic)** is partially addressed. The 6-background track is welcome, but the ordering change (CEEMDAN drops from 2nd to 5th) raises a question: were any of the synthetic failure-mode predictions (e.g., "STL fails under drift") contradicted in the semi-synthetic setting? A brief clarification would help confirm the benchmark's practical transfer value.
> >
> > I maintain my current score.

---

> > > ### Author Response · Authors · 2026-04-07
> > >
> > > We sincerely thank the reviewer for their interest in our work and the constructive questions.
> > >
> > > # Response to Q3
> > >
> > > Most synthetic conclusions are not overturned in the semi-synthetic setting. The main mechanism-level picture is preserved. semi-synthetic datasets are significantly more difficult than pure synthetic. The performance drop is systematic—driven by real background aliasing and intensity, not individual method fluctuations. The most notable shift is CEEMDAN dropping from 2nd to 5th, so why CEEMDAN's rank drops?
> > >
> > > * **Component Allocation Failure:** CEEMDAN's frequency tracking remains competitive (mean spectral correlation `0.769`, max-lag `0.704`), but its **Seasonal $R^2$ collapses to `-0.232`** (SSA, STL, MA remain positive). The penalty comes from amplitude/shape metrics, not spectral ones.
> > > * **Mode Mixing in Real Backgrounds:** Real-world series contain inherent low-frequency structures. CEEMDAN  (a mode decomposition method)  struggles to separate injected signals from these, detecting the frequency but incorrectly mixing background energy into the seasonal mode.
> > > * **Scale Sensitivity:** At `scale=0.5`, CEEMDAN's Seasonal $R^2$ remains strong (`0.57`–`0.68`). At `scale=1.0`, it crashes to strong negative values across all mechanisms.
> > >
> > > CEEMDAN does not lose the cycle entirely; strong real-world backgrounds induce severe mode mixing, destroying $R^2$ and pulling down its average rank.
> > >
> > > # Response to Q1
> > >
> > > We conducted a systematic probe into decomposition-aware modules extracted from deep forecasting architectures, evaluating them as standalone decomposers. We emphasize that rigorously answering **"how well do neural trend/seasonal extraction blocks actually perform?" is far from a straightforward question.** Extracting these blocks from their original architectures introduces fundamental mismatches that each demand dedicated investigation: (i) these blocks are trained under forecasting objectives, not reconstruction—adapting them to recover true signal components requires rethinking the training target; (ii) removing backpropagation pathways strips away learned interactions, yet retaining them demands redesigning loss functions to directly supervise component separation; and (iii) it remains open whether decomposition task can redesign via transfer learning or pretrained models. Each path constitutes a substantial research agenda likely requiring dedicated studies beyond a single paper. The present work implements the simplest, most assumption-free protocol—evaluating blocks as label-free mechanism-proxy operators to establish the controlled benchmark and classical-bias mapping as a prerequisite foundation. **We plan to pursue these deeper investigations in future work, and we welcome the broader community to engage with this direction, as rigorously understanding neural decomposition blocks can directly inform the design of next-generation time series architectures and foundation models.**
> > >
> > > These proxies form five families:
> > >
> > > | Family | Sources |
> > > |---|---|
> > > | **Smoothing/MA** | Autoformer, DLinear, xPatch (AAAI'25), LEDDAM (ICML'24) |
> > > | **Smoothing+Template** | InParformer (AAAI'23), DeLELSTM (IJCAI'23), AMD (AAAI'25), ST-MTM (KDD'25) |
> > > | **Harmonic/Frequency-Aware** | Parsimony (NeurIPS'24), Times2D (AAAI'25), FreqMoE (AISTATS'25), TimeKAN (ICLR'25) |
> > > | **Wavelet/Multiresolution** | WaveForM (AAAI'23), WaveletMixer (AAAI'25) |
> > > | **Learned-Basis Ref.** | N-BEATS Interpretable |
> > >
> > > There is no universal "neural block failure" mode. Neural blocks exhibit distinct behaviors mirroring classical methods.
> > >
> > > **1. Neural Blocks Learn Classical Biases:**
> > > * **Wavelet Replications:** `waveform_block` and `waveletmixer_block` failure modes nearly exactly mirror classical Wavelet (vulnerable to `single_sine`, `drift`, `piecewise`), proving they learned a classical wavelet bias rather than a novel one.
> > > * **Adaptive Spectral Learning:** The strongest frequency-aware blocks (`times2d`, `freqmoe`, `timekan`) behave like adaptive spectral methods (VMD/CEEMDAN), successfully handling drift and piecewise regimes via adaptive frequency tracking.
> > > * **Enhanced Smoothing:** Neural MA blocks (Autoformer/DLinear) resemble classical MA/STL but outperform classical MA on frequency drift, indicating learned components compensate for fixed-period limitations.
> > >
> > > **2. Diagnosing Neural Failures via Classical Successes:**
> > > * **Component Routing:** If classical MA/STL succeeds but a neural smoother fails, the issue is component routing—misassigning trend energy to the seasonal branch—not smoothing impossibility.
> > > * **Lack of Adaptivity:** If VMD/CEEMDAN succeeds on drift/piecewise but a neural block collapses, the block lacks local adaptivity regardless of parameter count.
> > >
> > > **3. High Intra-Family Heterogeneity:**
> > > Possessing a "learned prior" or "harmonic prior" does not guarantee successful decomposition. Methods within the same category exhibit vastly different failure modes (e.g., `times2d` fails differently from `delelstm`).

---

### Official Review · Reviewer_zQyK · 2026-03-11

**Soundness:** 3
**Presentation:** 3
**Significance:** 3
**Originality:** 3
**Overall Recommendation:** 4
**Confidence:** 4

**Summary:**

This paper proposes an diagnostic benchmark for time series decomposition by generating synthetic series with known ground truth trend, seasonality, and residual, then evaluating how well different decomposition methods recover these components. It defines a taxonomy of patterns, including nonstationary cases like drift and regime switching, wraps multiple existing decomposers under a unified interface, and uses component specific metrics to compare them. The main result is a diagnostic map of different model families to illustrate downstream interpretability value.

**Compliance With Llm Reviewing Policy:**

Affirmed.

**Final Justification:**

I think this work provides a comprehensive analysis and benchmarking on the specific topic of decomposition. My previous biggest concern is 1. the lack of real data, 2. what is the justification/impact of this benchmark. For the first question, the author made a point during rebuttal that addrees my concerns, but again, although I understand that real data is hard to obtain, using synthetic data is still a major shortness of this paper. For the second question, the authors provide a line of works which are mostly how decomposition can be used in current era. This partailly address their point. Although the actual impact of "accurate decomposition", which is defined by their benchmark, is still questionable (e.g., will it help other models that use decomposition?), but this is beyond the scope of this rebuttal

**Key Questions For Authors:**

- Can you add a broader qualitative real data evaluation using proxy checks with properties like known periodicities, stability under resampling, and robustness across time beyond a couple of case studies?
- Your alignment aggregates components by dominant frequency matching anchored to the injected primary period. How robust are results to alternative alignment rules that do not rely on the injected period? Does the alignment protocol favor certain families? Even one ablation would help rule out evaluation artifacts.

**Limitations:**

yes

**Strengths And Weaknesses:**

Strength:
- The synthetic benchmark is mechanism controlled, so it can isolate specific failure modes such as nonstationary seasonality and regime switching and provide more specific diagnostic insights.
- The evaluation uses multiple metrics and includes a small real data case study track to show potential usefulness beyond synthetic.

Weakness:
- Synthetic to real gap. Even realistic synthetic data can differ drastically from real applications, which might be much more diverse under many different contexts, so conclusions may not transfer without stronger real data support.
- The task of decomposition may feel less impactful today if large pretrained models can solve many time series tasks directly such as forecasting without manual feature engineering. So I think one important aspect that the paper should justify is the relevance of decomposition such as interpretability, discovery, etc.

---

> ### Author Rebuttal · Authors · 2026-03-30
>
> We appreciate the reviewer's focus on the synthetic-to-real gap and on possible evaluation artifacts. The original time series decomposition package was already included in the submission attachment. During review, we continued extending the benchmark and software stack, including an accelerated C++-backed faster Version.
>
> # 1. Decomposition is not just manual feature engineering for classical forecasting.
> At core, a decomposition benchmark studies how well different methods extract trend and periodic structure from time-series data. It is a basic problem in time-series research, because trend and periodic structure can serve as explicit outputs, embeddings, model components, or representation layers across forecasting-Autoformer (NeurIPS2021), FEDformer (ICML2022), DLinear (AAAI2023), Diffusion-TS (ICLR2024), mr-Diff (ICLR2024), TimeMixer(ICLR24), classification-MPTSNet(AAAI2025), anomaly detection-TimeCAD(ECML2023), and **even within recent foundation models for time series, decomposition-inspired techniques remain highly valuable. It continues to provide a useful structural prior for learning better tokens and richer temporal semantics WaveToken (ICML2025)**. And our own symbolic-regression track already also shows downstream value. In that sense, we view this benchmark not as an endpoint for classical visual decomposition, but as infrastructure for designing stronger ways to extract core trend and periodic representations in future models.
>
> # 2. Broader qualitative real-data evaluation with proxy checks.
>
> Before summarizing the new real-data results, we want to clarify the underlying constraint. It is not that we are unwilling to use real data; rather, real time series rarely come with genuine component-level ground truth for trend, seasonality, and residual. In practice, the strongest supervision available on real data is usually mechanism-informed *approximate* ground truth. This is exactly why the paper originally used two datasets with relatively clear physical structure, and why the rebuttal expands the real-data side with both more such cases and a broader proxy/stability panel.
>
> On real-data support, we now distinguish two evidence tracks:
>
> | Track | Evidence |
> | --- | --- |
> | Mechanism-aware cases | CO$_2$: SSA seasonal/trend $R^2 = 0.993/1.000$; tides: MSTL seasonal $R^2 = 0.956$, target-frequency coverage $1.000$, M2/S2 separation success $1.000$ |
> | Broader proxy panel | CH$_4$, NDVI, GPCC precipitation, Arctic sea ice, sunspots, and QBO; same 3 proxy criteria with 4 bootstrap repeats plus rolling-window checks; proxy/stability evidence rather than exact component recovery |
>
> On the Broader proxy panel, the aggregate results are:
>
> | Method | Band plaus. | Resampling stab. | Spectrum overlap |
> | --- | ---: | ---: | ---: |
> | SSA | $0.967$ | $0.651$ | $0.950$ |
> | MSTL | $0.918$ | $0.652$ | $0.951$ |
> | STL | $0.910$ | $0.639$ | $0.947$ |
> | VMD | $0.792$ | $0.570$ | $0.882$ |
> | CEEMDAN | $0.782$ | $0.557$ | $0.903$ |
> | Wavelet | $0.172$ | $0.666$ | $0.924$ |
>
> We will therefore present these additions as evidence that the synthetic conclusions remain plausible on real data, not as over-claimed exact recovery on real-world data.
>
> # 3. Alignment robustness beyond the injected period.
>
> We also added an alignment-robustness ablation that asks a simple question: if a method outputs several components, how much does its score depend on how those components are matched back to trend, seasonality, and residual during evaluation? On the same 5-scenario, 24-draw, length-512 synthetic core suite, we compare three matching rules: injected matching, unsupervised peak matching, and truth-guided matching. In the truth-guided column, ground truth is used only to choose the best matching for scoring; the decomposition outputs themselves are unchanged:
>
> | Method | Injected | Unsupervised | Truth-guided |
> | --- | ---: | ---: | ---: |
> | MSTL | $0.757$ | $0.757$ | $0.757$ |
> | STL | $0.756$ | $0.756$ | $0.756$ |
> | SSA | $0.730$ | $0.763$ | $0.862$ |
> | CEEMDAN | $0.727$ | $0.355$ | $0.867$ |
> | Wavelet | $0.636$ | $0.292$ | $0.720$ |
> | VMD | $-0.283$ | $0.274$ | $0.559$ |
>
> The point is therefore not to pick one universally ``correct'' alignment rule, but to show how much the reported ranking depends on that design choice. We will make the paper more careful about ranking statements for multi-component families and emphasize that the alignment rule is part of the evaluation design rather than a neutral afterthought.

---

> > ### Author Rebuttal · Reviewer_zQyK · 2026-04-02
> >
> > Thanks the author detailed response. This fully addresses my concerns.

---

> > > ### Author Response · Authors · 2026-04-07
> > >
> > > Dear Reviewer zQyK,
> > >
> > > Thank you very much for your positive feedback and for confirming that your concerns have been fully addressed!
> > >
> > >
> > > We deeply appreciate the time and care you put into reviewing our work, and we are incredibly grateful for your guidance and support!

---

### Official Review · Reviewer_WfuW · 2026-03-12

**Soundness:** 3
**Presentation:** 3
**Significance:** 3
**Originality:** 3
**Overall Recommendation:** 4
**Confidence:** 3

**Summary:**

This paper first defines time series decomposition as an independent task amenable to quantitative evaluation. It constructs a benchmarking framework capable of identifying underlying mechanisms.

**Compliance With Llm Reviewing Policy:**

Affirmed.

**Final Justification:**

My initial concerns primarily centered on the rigor of the experiment  in the paper. Based on the author’s response, I believe the suggested revisions will adequately address these issues. Therefore, I will give the paper a weak accept score. Since the concerns regarding the rigor of the experiment have been addressed, and taking into account the paper’s overall rigor, originality, significance, and clarity, I still award it a “weak accept” rating.

**Key Questions For Authors:**

1.More deeplearning method can make the benchmark more solid.

2.Providing a parameter-tuned version of the parameter-sensitive algorithm can further enhance its persuasiveness.

3.Does the core advantage of the “mechanism-recognizable” benchmark mentioned in the paper, compared to previous temporal decomposition datasets (such as subtasks of the UCR/UEA dataset), lie solely in the “known true components”?

4.The default experimental setting in the paper causes certain classic methods (such as VMD) to perform very poorly, which may not true.  If a reasonable budget were allocated for hyperparameter tuning of VMD, SSA, EMD and so on, it may  change the ranking of their performance.  Differences in how each method handles the beginning part and end part of a time series are not elaborated upon. for instance, it is unclear whether boundary artefacts in Wavelet or EMD have a significant impact on metrics under small sample size setting. In practice, there are often thousands or even tens of thousands of correlated sequences .Wether the benchmark be extended to multi-sequence decomposition methods.

**Limitations:**

yes. The author acknowledges that the issues of trend and seasonal have not yet been investigated.

**Strengths And Weaknesses:**

Strengths:

In the past, nearly all work treated decomposition as a subsidiary step of forecasting, revealing the failure modes of existing methods. Treating time series decomposition as a core task in its own right and constructing a controllable suite of synthetic benchmarks. While previous research has largely treated decomposition as a preprocessing step prior to forecasting, the authors take a more forward-looking approach, establishing a common evaluation platform for subsequent algorithmic research and applications.

This study summarizes the effects of assumptions such as the fixed-period assumption, the low-rank subspace assumption, and the time-frequency localization assumption on decomposition performance, providing a concise and clear reference for selecting appropriate models in practical scenarios.

The paper is well-written, is easy to understand.

Weakness:

This bench lack of some deeplearning method.

The default experimental setting in the paper causes certain classic methods (such as VMD) to perform very poorly, which may not true.  If a reasonable budget were allocated for hyperparameter tuning of VMD, SSA, EMD and so on, it may  change the ranking of their performance.  Differences in how each method handles the beginning part and end part of a time series are not elaborated upon. for instance, it is unclear whether boundary artefacts in Wavelet or EMD have a significant impact on metrics under small sample size setting. In practice, there are often thousands or even tens of thousands of correlated sequences .Wether the benchmark be extended to multi-sequence decomposition methods.

---

> ### Author Rebuttal · Authors · 2026-03-30
>
> We thank the reviewer for emphasizing fairness, boundary effects, modern baselines, and multi-sequence settings. The original time series decomposition package was already included in the submission attachment. During review, we continued extending the benchmark and software stack, including an accelerated C++-backed faster Version. We view this benchmark as maintained infrastructure for studying how models extract core trend and periodic structure inside newer architectures. The new evidence addresses these concerns.
>
> # 1. More deep-learning methods.
>
> We also want to clarify why the initial submission did not include a large neural-decomposition leaderboard. To the best of our knowledge, there is currently no dedicated deep-learning family whose primary purpose is standalone time-series decomposition and that is benchmarked as such in the same way that STL is decomposition method. If the reviewer is aware of a direct counterexample, we would be happy to discuss it in revision. Recent neural work instead uses decomposition more often as an internal block, module, or representation layer for extracting trend, seasonality, or multiscale structure inside
> larger forecasting models (**examples in rebuttal of zQyK 1.**) . Under the same protocol, we added the following neural experiments inlcude **Autoformer block** experiments and a backprop-trained learned-prior -interpretable **N-BEATS block**, with explicit trend and seasonality stacks.
> (**Full results in rebuttal of Reviewer FFgj 1.**)
>
>
>
> # 2. Parameter-tuned version.
>
> On fairness for parameter-sensitive methods, we added bounded tuning reruns under a train-only 60/20/20 split and the same bounded common budget. The fixed sweeps use VMD `K={2,3,4,5,6}`, `alpha={200,500,1000,2000,5000}`, and `init={1,2}`;  CEEMDAN `trials={50,100}` and `noise_scale={0.05,0.1,0.2}`; and SSA `grouping_rule={current_period_aware,spectral_auto}`, `rank={10}`, and `window={N/4,N/3,N/2,2N/3,P,2P,4P}`. To keep all parameter-sensitive methods on one footing, we summarize this check by the aggregate ordering of the 6 main families rather than by a mixed per-metric table.
> | Setting | Aggregate order (best -> worst) |
> | --- | --- |
> | Default | SSA > CEEMDAN > MA > STL > Wavelet > VMD |
> | Tuned | SSA > CEEMDAN > MA > VMD > STL > Wavelet|
>
> So the ranking change under common-budget tuning is modest rather than structural: CEEMDAN , SSA remains same, and VMD improves only from last to fouth. We will present tuned and default results side by side as a common-budget comparison rather than exhaustive per-dataset optimization.
>
> # 3. Does the core advantage lie solely in the known true components?
>
> First, UCR/UEA are time-series classification collections rather than decomposition datasets, and to our knowledge they do not provide the kind of component-level decomposition subtasks implied in the review. More importantly, the advantage of our benchmark is not only that true components are known. Known components enable exact recovery metrics, but the broader value is that we can vary one mechanism at a time and observe which inductive bias fails under which controlled stress. In the current core suite, this controlled design covers 5 named mechanism families at 24 draws per scenario and length 512, including single-period, multi-harmonic, multi-seasonal, drifting-frequency, and piecewise regime/event settings. The key value is therefore the combination of known components with controllable mechanism families and stress tests: fixed-period vs. drifting-frequency seasonality, single- vs. multi-seasonality, smooth vs. piecewise trends, and regime/event perturbations. This turns the benchmark into a diagnostic test of inductive-bias mismatch rather than only a labeled dataset. We will revise the paper to make that point explicit.
>
> # 4. Boundary artefacts and multi-sequence extension.
>
> We summarize the new boundary and multi-sequence evidence below.
>
> | Method | Mean abs. suite-score delta |
> | --- | ---: |
> | VMD | 1.097 |
> | Wavelet | 0.734 |
> | SSA | 0.311 |
> | CEEMDAN | 0.230$ |
> | STL | $0.224$ |
> | MSTL | $0.209$ |
>
> This boundary sweep uses lengths 128/256/512/1024 and reports the mean absolute synthetic suite-score delta between full-window and trimmed scoring. Edge handling matters most for VMD and wavelet, but is not wavelet-only.
>
> | Item | Result |
> | --- | --- |
> | Panel setup | Synthetic only; $C=8$ and $C=32$; 12 panels per setting; shared annual/semiannual, drifting-frequency, and regime-shift-with-local-shocks families |
> | MSSA overall | Mean channel trend/seasonal $R^2 = 0.989/0.970$; seasonal covariance error $0.035$; shared-loading correlation $0.999$ |
> | Hardest panel | Shared drifting-frequency panel with 32 channels |
> | MSSA vs baselines | MSSA $0.991/1.000$ vs STL $0.755/0.963$ vs single-sequence SSA $0.875/0.996$ |
>
> We therefore position this as strong initial evidence that the benchmark can extend beyond univariate decomposition, though it is not yet a full real-panel benchmark.

---

> > ### Author Rebuttal · Reviewer_WfuW · 2026-04-01
> >
> > Thanks. My concerns have been addressed. I maintain my score.

---

> > > ### Author Response · Authors · 2026-04-07
> > >
> > > We thank Reviewer WfuW for the constructive feedback.
> > >
> > > To clarify the Partially Resolved concerns, we conducted a systematic probe into decomposition-aware modules extracted from deep forecasting architectures, evaluating them as standalone decomposers. We emphasize that rigorously answering **"how well do neural trend/seasonal extraction blocks actually perform?" is far from a straightforward question.** Extracting these blocks from their original architectures introduces fundamental mismatches that each demand dedicated investigation: (i) these blocks are trained under forecasting objectives, not reconstruction—adapting them to recover true signal components requires rethinking the training target; (ii) removing backpropagation pathways strips away learned interactions, yet retaining them demands redesigning loss functions to directly supervise component separation; and (iii) it remains open whether decomposition task can redesign via transfer learning or pretrained models. Each path constitutes a substantial research agenda likely requiring dedicated studies beyond a single paper. The present work implements the simplest, most assumption-free protocol—evaluating blocks as label-free mechanism-proxy operators to establish the controlled benchmark and classical-bias mapping as a prerequisite foundation. **We plan to pursue these deeper investigations in future work, and we welcome the broader community to engage with this direction, as rigorously understanding neural decomposition blocks can directly inform the design of next-generation time series architectures and foundation models.**
> > >
> > > These proxies form five families:
> > >
> > > | Family | Sources |
> > > |---|---|
> > > | **Smoothing/MA** | Autoformer, DLinear, xPatch (AAAI'25), LEDDAM (ICML'24) |
> > > | **Smoothing+Template** | InParformer (AAAI'23), DeLELSTM (IJCAI'23), AMD (AAAI'25), ST-MTM (KDD'25) |
> > > | **Harmonic/Frequency-Aware** | Parsimony (NeurIPS'24), Times2D (AAAI'25), FreqMoE (AISTATS'25), TimeKAN (ICLR'25) |
> > > | **Wavelet/Multiresolution** | WaveForM (AAAI'23), WaveletMixer (AAAI'25) |
> > > | **Learned-Basis Ref.** | N-BEATS Interpretable |
> > >
> > > There is no universal "neural block failure" mode. Neural blocks exhibit distinct behaviors mirroring classical methods.
> > >
> > > **1. Neural Blocks Learn Classical Biases:**
> > > * **Wavelet Replications:** `waveform_block` and `waveletmixer_block` failure modes nearly exactly mirror classical Wavelet (vulnerable to `single_sine`, `drift`, `piecewise`), proving they learned a classical wavelet bias rather than a novel one. Specifically, `waveform_block` and `waveletmixer_block` largely inherit the brittle failure map of classical wavelets, especially on `single_sine`, `drift`, and `piecewise`—demonstrating that these neural blocks have not learned a novel inductive bias, but rather replicated the classical wavelet bias.
> > > * **Adaptive Spectral Learning:** The strongest frequency-aware blocks (`times2d`, `freqmoe`, `timekan`) behave like adaptive spectral methods (VMD/CEEMDAN), successfully handling drift and piecewise regimes via adaptive frequency tracking.
> > > * **Enhanced Smoothing:** Neural MA blocks (Autoformer/DLinear) resemble classical MA/STL but outperform classical MA on frequency drift, indicating learned components compensate for fixed-period limitations. Notably, the most consistent shared failure of `autoformer_block`, `dlinear_block`, and `moving_average_decomposition_block` is on `logistic_trend_multi_seasonal`: trend recovery is strong but seasonal recovery is poor, suggesting they function more as strong trend extractors than as true multi-seasonal decomposers.
> > >
> > > **2. Diagnosing Neural Failures via Classical Successes:**
> > > * **Component Routing:** If classical MA/STL succeeds but a neural smoother fails, the issue is component routing—misassigning trend energy to the seasonal branch—not smoothing impossibility.
> > > * **Lack of Adaptivity:** If VMD/CEEMDAN succeeds on drift/piecewise but a neural block collapses, the block lacks local adaptivity regardless of parameter count.
> > >
> > > **3. High Intra-Family Heterogeneity:**
> > > Possessing a "learned prior" or "harmonic prior" does not guarantee successful decomposition. Methods within the same category exhibit vastly different failure modes (e.g., `times2d` fails differently from `delelstm`). The key driver of large intra-family differences often lies not in the shared trend smoother, but in the representational capacity and parameterization of the seasonal branch.

---

### Decision · Program_Chairs · 2026-04-30

**Decision:**

Accept (regular)

**Comment:**

The paper introduces  a new benchmarking framework for time series decomposition based on synthetic data generation. The reviewers acknowledged a number of strengths such as clarity of presentation, introduction of new meaningful metrics, and cotrollability of generative mechanism. During the rebuttal phase, the authors  addressed all reviewers concerns on the rigor of the experiments, lack of real data and deep learning baselines and potential impact of this benchmark. Overall this is paper that make a meaningful contribution to the filed of time series analysis and decomposition and the therefore I recommend its acceptance.